# The co-receptor Tetraspanin12 directly captures Norrin to promote ligand-specific β-catenin signaling

**Elise S Bruguera\*, Jacob P Mahoney, William I Weis†**

Departments of Molecular & Cellular Physiology and Structural Biology, Stanford University School of Medicine, Stanford, United States

## eLife Assessment

This is a **fundamental** study that addresses the key question of how the tetraspanin Tspan12 functions biochemically as a co-receptor for Norrin to initiate β-catenin signaling. The strength of the work lies in the rigorous and **compelling** binding analyses involving various purified receptors, co-receptors, and ligands, as well as molecular modeling by AlphaFold that was subsequently validated by an extensive series of mutagenesis experiments. The study advances the field by providing a novel mechanism of co-receptor function and shedding new light on how signaling specificity is achieved in the complex Wnt/Norrin signaling system.

**\*For correspondence:**
ebruguer@stanford.edu

†Deceased

Competing interest: The authors declare that no competing interests exist.

**Abstract** Wnt/β-catenin signaling directs animal development and tissue renewal in a tightly controlled, cell- and tissue-specific manner. In the mammalian central nervous system, the atypical ligand Norrin controls angiogenesis and maintenance of the blood-brain barrier and blood-retina barrier through the Wnt/β-catenin pathway. Like Wnt, Norrin activates signaling by binding and heterodimerizing the receptors Frizzled (Fzd) and low-density lipoprotein receptor-related protein 5 or 6 (LRP5/6), leading to membrane recruitment of the intracellular transducer Dishevelled (Dvl) and ultimately stabilizing the transcriptional coactivator β-catenin. Unlike Wnt, the cystine knot ligand Norrin only signals through Fzd4 and additionally requires the co-receptor Tetraspanin12 (Tspan12); however, the mechanism underlying Tspan12-mediated signal enhancement is unclear. It has been proposed that Tspan12 integrates into the Norrin-Fzd4 complex to enhance Norrin-Fzd4 affinity or otherwise allosterically modulate Fzd4 signaling. Here, we measure direct, high-affinity binding between purified Norrin and Tspan12 in a lipid environment and use AlphaFold models to interrogate this interaction interface. We find that Tspan12 and Fzd4 can simultaneously bind Norrin and that a pre-formed Tspan12/Fzd4 heterodimer, as well as cells co-expressing Tspan12 and Fzd4, more efficiently capture low concentrations of Norrin than Fzd4 alone. We also show that Tspan12 competes with both heparan sulfate proteoglycans and LRP6 for Norrin binding and that Tspan12 does not impact Fzd4-Dvl affinity in the presence or absence of Norrin. Our findings suggest that Tspan12 does not allosterically enhance Fzd4 binding to Norrin or Dvl, but instead functions to directly capture Norrin upstream of signaling.

## Introduction

Wnt/β-catenin signaling is essential to metazoan development, tissue homeostasis, and regeneration. Wnts are secreted growth factors that act through simultaneous binding to, and heterodimerization of, two membrane co-receptors: Frizzled (Fzd1-10 in humans) and low-density lipoprotein receptor-related protein 5 or 6 (LRP5/6) (*Bourhis et al., 2010*; *Janda et al., 2017*; *Tsutsumi et al., 2023*). Fzd

recruits the cytoplasmic protein Dishevelled (Dvl1/2/3 in humans) (*Ma et al., 2020*; *Mahoney et al., 2022*; *Tauriello et al., 2012*; *Yang-Snyder et al., 1996*), which is necessary to recruit and inhibit the proteins responsible for the constitutive degradation of β-catenin (*Cliffe et al., 2003*; *Piao et al., 2008*; *Schwarz-Romond et al., 2007*; *Stamos et al., 2013*; *Zeng et al., 2008*). Wnt morphogens thereby stabilize β-catenin to induce signaling. β-Catenin directs the transcription of genes that drive cell division, migration, and differentiation during development as well as stem cell maintenance and renewal throughout the lifecycle (*Rim et al., 2022*). Consequently, dysregulated β-catenin signaling can lead to cancer and degenerative diseases (*Nusse and Clevers, 2017*).

The secreted ligand Norrin, which is structurally distinct from Wnt, binds to Fzd4 and LRP5/6 to activate β-catenin signaling (*Chang et al., 2015*; *Ke et al., 2013*; *Xu et al., 2004*). Norrin is a disulfide-linked dimer with a transforming growth factor-β-like cystine knot fold that selectively binds the Fzd4 cysteine-rich domain (CRD) with high affinity, as well as LRP5/6 β-propeller-EGF repeats 1 and 2 (E1E2) and heparan sulfate proteoglycans (HSPGs) (*Chang et al., 2015*; *Ke et al., 2013*; *Xu et al., 2004*). Norrin is specifically required for retinal vascularization and blood-retina barrier (BRB) integrity as well as blood-brain barrier (BBB) integrity in the cerebellum (*Wang et al., 2012*; *Ye et al., 2009*), where it is partially redundant with Wnt7a/7b signaling (*Wang et al., 2018*).

With 10 different Fzd subtypes and 19 different Wnts, along with Norrin, in the human genome, exquisite spatial and temporal signaling specificity is achieved through differential expression patterns (*Nusse and Varmus, 1992*) and pairwise affinities (*Dijksterhuis et al., 2015*) of the receptors and ligands. Specificity is also achieved through the expression of accessory proteins or co-receptors that modulate the expression levels, activity, or localization of core pathway components (*Cruciat and Niehrs, 2013*; *Schulte, 2015*). For example, the adhesion G protein-coupled receptor Gpr124 and GPI-anchored cell-surface metalloprotease inhibitor RECK were recently found to selectively enhance Wnt7a/7b signaling through a direct RECK-Wnt7a/7b interaction to control angiogenesis and maintenance of the BBB in the central nervous system (*Cho et al., 2017*; *Eubelen et al., 2018*; *Vallon et al., 2018*; *Vanhollebeke et al., 2015*). Similarly, the tetraspanin Tspan12 is required for Norrin-directed retinal angiogenesis and maintenance of the BRB (*Junge et al., 2009*). Mutations in Norrin, Fzd4, LRP5, β-catenin, and Tspan12 are associated with inherited diseases of the retinal vasculature, most notably familial exudative vitreoretinopathy (FEVR) (*Jiao et al., 2004*; *Nikopoulos et al., 2010*; *Panagiotou et al., 2017*; *Poulter et al., 2010*; *Robitaille et al., 2002*; *Shastry et al., 1997*; *Toomes et al., 2004*), and therapeutic targeting of Norrin/β-catenin signaling shows promise for modulating the BRB and BBB (*Chidiac et al., 2021*; *Ding et al., 2023*; *Nguyen et al., 2022*; *O'Brien et al., 2023*; *Phoenix et al., 2016*). However, the mechanism by which Tspan12 promotes Norrin signaling is unclear. A molecular understanding of how Tspan12 and other modulators achieve their function would enable the development of therapeutics targeting the Wnt/β-catenin pathway with greater specificity.

Members of the tetraspanin family cluster into nanodomains in the plasma membrane and modulate diverse cellular processes by binding to other membrane proteins to influence their trafficking, localization, conformation, and ligand recruitment (*Mattila et al., 2013*; *Sugiyama et al., 2023*; *Susa et al., 2024*; *van Deventer et al., 2017*). Structurally, tetraspanins have four transmembrane domains along with a small extracellular loop (SEL) and a large extracellular loop (LEL), the latter of which is implicated in many protein-protein interactions (*Susa et al., 2024*). Tetraspanins can also recruit kinases intracellularly (*Lapalombella et al., 2012*; *Zhang et al., 2001*). These observations provide potential routes by which Tspan12 might influence Norrin/β-catenin signaling. Indeed, Tspan12 has been shown to co-traffic, co-localize, and co-immunoprecipitate (co-IP) with Fzd4 and Norrin, all of which require the Tspan12 LEL (*Junge et al., 2009*; *Lai et al., 2017*), and purified Tspan12 LEL, grafted onto an antibody, captured Norrin from conditioned media (*Hsieh and Chang, 2021*). In addition, bioluminescence resonance energy transfer experiments suggest that Tspan12 forms Tspan12-Tpsan12 homodimers and Tspan12-Fzd4 heterodimers in cells (*Ke et al., 2013*). Tspan12 restores signaling when the Norrin-Fzd4 interface is weakened by point mutations (*Lai et al., 2017*), suggesting that Tspan12 might enhance Norrin-Fzd4 binding. However, it is unclear whether Tspan12 achieves this function through a direct interaction with either Fzd4 or Norrin. Furthermore, whether or how Tspan12 works with the core Fzd4 and LRP5/6 co-receptor pair to transduce Norrin signaling is unknown.

Here, we use purified Norrin and purified receptors reconstituted in lipid nanodiscs to explore the mechanism of Tspan12 action in Norrin/β-catenin signaling. We demonstrate that Tspan12 directly

binds Norrin with high affinity and show that Norrin-Tspan12 binding is compatible with Norrin-Fzd4 binding, but not LRP6 or HSPG binding. Our work supports a model in which Tspan12-expressing cells preferentially capture Norrin, which is then handed off to Fzd4 for association with LRP5/6 and consequent signaling.

## Results

### Tspan12 binds Norrin, but not Fzd4, with high affinity

We first sought to determine whether Tspan12 directly binds Norrin. We purified Tspan12, reconstituted it into lipid nanodiscs (*Figure 1—figure supplement 1*), and found that Tspan12 directly binds purified Norrin with an affinity of 10.4±1.2 nM as measured by biolayer interferometry (BLI) (*Figure 1A–D* and *Figure 1—figure supplement 2A*). This interaction is stronger when Tspan12 is in a lipid environment than when it is in a glyco-diosgenin (GDN) detergent micelle (*Figure 1—figure supplement 2B*). Full-length Tspan12 and the purified LEL bind Norrin with similar affinity (*Figure 1E*). A purified chimera of Tspan12 with the LEL replaced by that of Tspan11, which does not enhance Norrin signaling (*Lai et al., 2017*), does not bind Norrin (*Figure 1F*). Together, these findings demonstrate that direct Norrin-Tspan12 binding is mediated by the Tspan12 LEL.

We next investigated the putative Fzd4-Tspan12 interaction, which has been shown by co-IP to depend on the Tspan12 LEL (*Lai et al., 2017*). We did not detect binding between the purified LEL and nanodisc-embedded Fzd4 (*Figure 1—figure supplement 2C*) nor between the Fzd4 extracellular domain (the CRD along with the linker to the transmembrane domains, termed 'CRDL') and nanodisc-embedded Tspan12 (*Figure 1—figure supplement 2D*). However, this does not definitively rule out a direct interaction between the two receptors, as it may be weak and require a two-dimensional (2D) membrane environment and/or be mediated, at least in part, by the transmembrane domains.

### Identification of the Norrin-Tspan12 binding site

To determine whether the Norrin-Tspan12 interaction would be spatially compatible with Norrin binding to other co-receptors, we sought structural information on the Norrin-Tspan12 binding interface. While individual crystal structures of Norrin alone and Norrin bound to both the Fzd4 CRD and the heparin mimic sucrose octasulfate (SOS) have been solved (*Chang et al., 2015*; *Ke et al., 2013*; *Shen et al., 2015*), no experimental structure of Tspan12 or Norrin-bound Tspan12 exists. We used AlphaFold (*Evans et al., 2022*; *Jumper et al., 2021*) to predict the structure of Tspan12 alone (*Figure 2—figure supplement 1A–C*), bound to a single protomer of Norrin (*Figure 2A*, *Figure 2—figure supplement 1D and E*), and bound to a Norrin dimer (*Figure 2C*, *Figure 2—figure supplement 1F and G*). As expected, the predicted structure of Norrin within the AlphaFold models is identical to the experimentally determined structure of Norrin (*Figure 2—figure supplement 1H*). The Tspan12 LEL structure is nearly identical between the three predicted models, irrespective of Norrin binding (*Figure 2—figure supplement 1I–K*). However, the three models of Tspan12 vary slightly in the position of the SEL and LEL relative to the transmembrane domains (*Figure 2—figure supplement 1I*), which we attribute to uncertainty in the prediction: the SEL and the extracellular ends of all four transmembrane domains, which link the SEL and LEL to the transmembrane domains and therefore control their positioning, exhibit lower predicted local distance difference test (pLDDT) scores, indicating lower confidence in their predicted positions (*Figure 2—figure supplement 1B, D, and F*). This uncertainty is consistent with the variability in conformations adopted by tetraspanin transmembrane domains, which have been solved with a cavity between the TMs opening extracellularly (*Lipper et al., 2023*; *Umeda et al., 2020*; *Yang et al., 2020*; *Zimmerman et al., 2016*) or with more tightly packed transmembrane domains (*El Mazouni and Gros, 2022*; *Susa et al., 2021*; *Yanagisawa et al., 2023*). The transition between these states has also been observed in molecular dynamics simulations (*Zimmerman et al., 2016*).

We were most interested in the predicted structure of the Norrin-binding domain of Tspan12, the LEL. Tetraspanin LELs contain five helices, A–E (*Figure 2—figure supplement 1J*), and are stabilized by 2, 3, or 4 disulfide bonds (*Susa et al., 2024*). The predicted Tspan12 LEL 'stalk' helices A and B resemble helices A and B within experimental structures of other LEL domains, such as CD81 (*Zimmerman et al., 2016*), Uroplakin 1A (*Yanagisawa et al., 2023*), and Peripherin 2 (*El Mazouni and Gros, 2022*; *Figure 2—figure supplement 1L–N*). The hypervariable 'head' region comprised

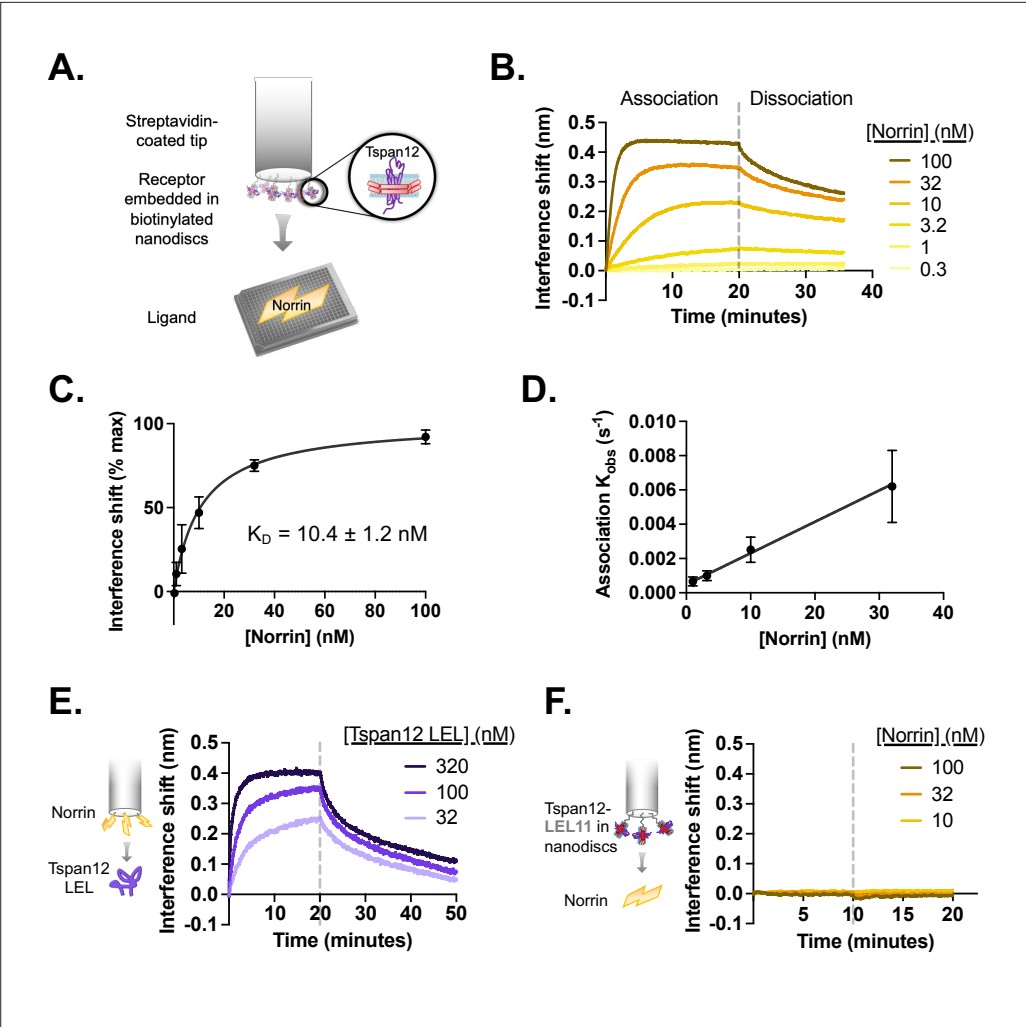

**Figure 1.** Tspan12 binds directly to Norrin with nanomolar affinity via the large extracellular loop (LEL). (**A**) Schematic of biolayer interferometry (BLI) setup for Tspan12-Norrin binding: Tspan12 lacking the C-terminal tail (ΔC), inserted into biotinylated nanodiscs, is immobilized on a streptavidin-coated biosensor, and Norrin association and dissociation are monitored in real time. (**B**) BLI traces of Norrin at indicated concentrations binding to and dissociating from Tspan12. (**C**) Steady-state binding curve fit to Norrin-Tspan12 binding (mean ± SD from three independent replicates at each concentration of Norrin) gives a $K_D$ of 10.4±1.2 nM (mean ± SEM). (**D**) Observed association rate constant ($K_{obs}$), determined from fitting BLI association traces (mean ± SD in three independent experiments), is linearly dependent on Norrin concentration with a slope $K_{on} = 0.00019 ± 0.00003$ nM$^{-1}$ s$^{-1}$ (mean ± SEM). When combined with the $K_{off} = 0.0014 ± 0.00016$ s$^{-1}$ (mean ± SEM) determined from fitting the dissociation traces, we obtain a kinetic $K_D$ of 7.4±1.4 nM (mean ± SEM). (**E**) BLI traces of the soluble MBP-tagged Tspan12 LEL domain, at the indicated concentrations, associating to and dissociating from a biosensor loaded with MBP-tagged Norrin. Kinetic fitting gives an apparent affinity of 16±3 nM (mean ± SEM). (**F**) BLI traces of 10, 32, or 100 nM Norrin show no binding to a biosensor loaded with a nanodisc-embedded chimeric Tspan12 with the LEL replaced by that of Tspan11.

The online version of this article includes the following source data and figure supplement(s) for figure 1:

**Source data 1.** Steady-state interference shift and $K_{obs}$ values used to generate *Figure 1C and D*.

**Figure supplement 1.** Tspan12 purification.

**Figure supplement 1—source data 1.** Original files of gels in *Figure 1—figure supplement 1*.

**Figure supplement 1—source data 2.** Labeled gels in *Figure 1—figure supplement 1*.

**Figure supplement 2.** Tspan12 and Fzd4 each bind Norrin, but not one another, with high affinity through their extracellular domains.

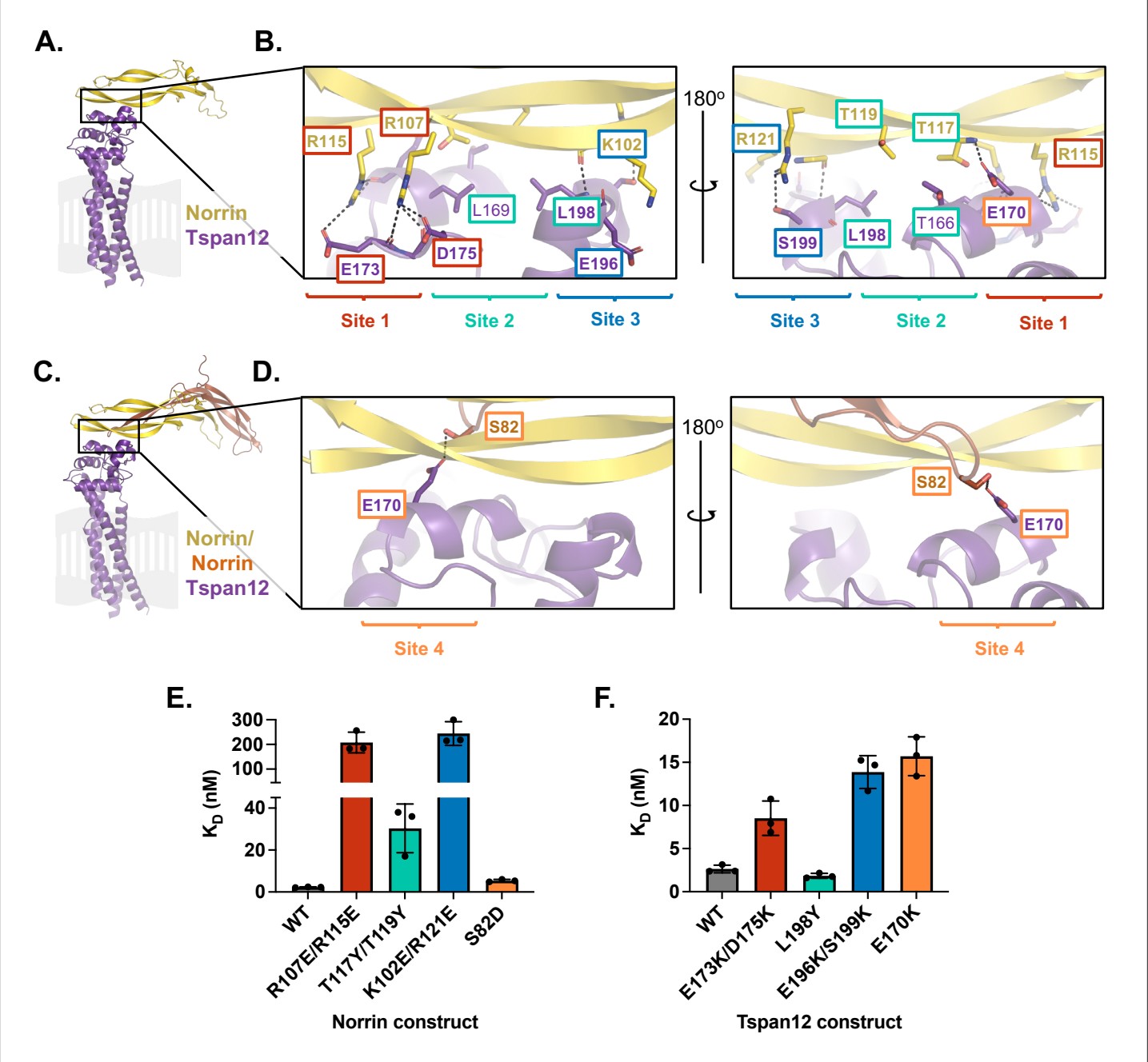

**Figure 2.** Mutational analysis of AlphaFold-predicted Tspan12-Norrin binding site. (**A**) AlphaFold model of one Norrin protomer (yellow) bound to Tspan12 (purple), with the expected location of the plasma membrane shown in gray. (**B**) Zoomed view of the predicted Tspan12/Norrin binding site, front and rear view (flipped 180°). Predicted polar interactions are indicated with dark gray dashed lines. Within the binding interface, Site 1 (red), Site 2 (teal), and Site 3 (blue) are indicated. Bold residue labels indicate residues mutated for binding assays. (**C**) AlphaFold model of Tspan12 bound to Norrin dimer and (**D**) zoomed view of indicated area, showing the predicted polar interaction between residue S82 on the second Norrin protomer (orange) and residue E170 on Tspan12, termed Site 4 (orange). (**E**) Binding affinities (mean ± SD) for the indicated Norrin mutants binding to full-length wild-type (WT) Tspan12 and (**F**) WT Norrin binding to the indicated Tspan12 mutants, calculated from association and dissociation fits to biolayer interferometry (BLI) traces of 32 nM Norrin binding to Tspan12 in triplicate (see *Figure 2—figure supplement 3A and B*). Colors correspond to sites within the binding interface. Kinetic traces and kinetic constants are shown in *Figure 2—figure supplement 3*, and affinities and kinetic constants are reported in *Supplementary file 1*.

The online version of this article includes the following source data and figure supplement(s) for figure 2:

**Source data 1.** Norrin-Tspan12 mutant affinities used to generate *Figure 2E and F*.

*Figure 2 continued on next page*

*Figure 2 continued*

**Figure supplement 1.** AlphaFold structural prediction of Tspan12 bound to Norrin.

**Figure supplement 2.** Purification of Norrin and Tspan12 mutants.

**Figure supplement 2—source data 1.** Original files of gels in *Figure 2—figure supplement 2*.

**Figure supplement 2—source data 2.** Labeled gels in *Figure 2—figure supplement 2*.

**Figure supplement 3.** Binding kinetics of Norrin and Tspan12 mutants.

**Figure supplement 4.** Binding affinity and kinetics of mutant Norrin binding to mutant Tspan12.

of helices C and D align less well to other tetraspanins, as does helix E, where residue positions were predicted with lower pLDDT confidence scores.

The AlphaFold model predicts a highly acidic patch on the C and D helices of the Tspan12 LEL as the binding site for a highly basic patch on Norrin β strands 5 and 6 (*Figure 2A*, *Figure 2—figure supplement 1H–K and O–P*), burying about 1200 Å$^2$ of surface area. Salt bridges are predicted between Norrin residues R107 and R115 and Tspan12 residues E175 and D173, respectively; we have termed this region Site 1 (*Figure 2B*). The model also predicts a region composed of hydrophobic contacts including Norrin residues T119 and T117 and Tspan12 residues T166, L169, and L198, which we have termed Site 2. Site 3 consists of electrostatic interactions between Norrin residues K102 and R121 and Tspan12 residues E196 and S199, as well as the C-terminal end of the Tspan12 helix D dipole. If two Norrin protomers are included in the Norrin-Tspan12 AlphaFold Multimer prediction (*Figure 2C*), an additional contact is predicted between E170 on Tspan12 and S82 on the second copy of Norrin, termed Site 4 (*Figure 2D*).

To experimentally test this structural prediction, charge-reversal mutations were introduced at side chains involved in polar interactions within Sites 1, 3, and 4; at Site 2, smaller buried side chains were mutated to bulkier residues (*Figure 2—figure supplement 2*). The binding of purified mutants was quantified using BLI by fitting the association and dissociation traces of Norrin at a single concentration (32 nM) to obtain kinetic binding constants (*Figure 2—figure supplement 3*, *Figure 2—figure supplement 4*, and *Supplementary file 1*). By this metric, wild-type (WT) Norrin binds full-length WT Tspan12 with an apparent affinity of 2.5±0.2 nM (mean ± SEM). Relative to this benchmark, mutations within Site 1 diminished Norrin-Tspan12 binding: the Norrin double mutant R107E/R115E binds WT Tspan12 with 84-fold weaker affinity, and WT Norrin binds Tspan12 E173K/D175K with 3.4-fold weaker affinity (*Figure 2E–F*, red bars). Given that these charge-swapping mutations were made at sites predicted to interact, we tested the binding of mutant Norrin R107E/R115E to mutant Tspan12 E173K/D175K, with the hypothesis that these mutations might be compensatory. However, the two mutants did not bind one another at the concentration of Norrin tested (*Figure 2—figure supplement 4*).

Mutations to hydrophobic contacts at Site 2 have more modest effects on Tspan12-Norrin affinity. The conservative mutation L198Y in Tspan12 does not appreciably change its binding affinity for Norrin, while the double mutation T117Y/T119Y in Norrin decreases its affinity for WT Tspan12 12-fold (*Figure 2E–F*, green bars). A T119P mutation in Norrin is associated with Norrie disease in humans, but this mutation may destabilize Norrin structure in addition to disrupting the Norrin-Tspan12 binding interface.

The most dramatic effects on Tspan12-Norrin binding were observed when mutations were made at Site 3. The mutations K102E/R121E together decrease Norrin's affinity for WT Tspan12 by two orders of magnitude, and WT Norrin binds Tspan12 E196K/S199K with sixfold weaker affinity than WT Tspan12 (*Figure 2E–F*, blue bars). However, these two mutants (Norrin K102E/R121E and Tspan12 E196K/S199K) bind each other with an affinity of 1.3±0.06 nM (*Figure 2—figure supplement 4*), slightly stronger than the WT-WT interaction. The fact that these charge-reversal mutations can compensate for one another supports the structural prediction that these two regions of Norrin and Tspan12 physically interact. Of note, a disease-associated Norrin mutation at this site, R121W, impairs signaling in a cell-based assay (*Chang et al., 2015*), which we propose may be attributed to deficient Tspan12 binding.

At Site 4, Norrin S82D binds WT Tspan12 with twofold lower affinity than WT, and WT Norrin binds Tspan12 E170K with sixfold lower affinity than WT (*Figure 2E–F*, orange bars). The two charge-swapped mutants Norrin S82D and Tspan12 E170K bind with an affinity of 5.2±0.7 nM (*Figure 2—figure*

*supplement 4*); thus, the mutation in Norrin at this site compensates for the deleterious effects of the mutation in Tspan12 at this site. Again, this suggests that these two residues are in proximity in the Norrin-Tspan12 complex, as in the AlphaFold-predicted model.

AlphaFold predictions vary in their accuracy even when reported with high confidence scores, with errors ranging from local backbone and side chain distortions to shifts in the orientation of entire domains (*Terwilliger et al., 2024*). Our ability to weaken Tspan12-Norrin affinity with mutations at the predicted interface suggests that the predicted interface is correct but does not rule out the possibility that the relative orientations of Tspan12 and Norrin within the model are erroneous. At Sites 3 and 4, charge-reversal mutations on Norrin can compensate for charge-reversal mutations on Tspan12 (*Figure 2—figure supplement 4* and *Supplementary file 1*), in strong support of direct interactions between the predicted mutated sites. However, this was not the case for Site 1. Nevertheless, this structural model allowed us to generate testable hypotheses regarding whether Norrin could bind Tspan12 alongside its other binding partners, which we next sought to test experimentally.

## Norrin can bind Tspan12 simultaneously with Fzd4, but not HSPGs

When the predicted Norrin-Tspan12 structure is compared with the available crystal structure of Norrin bound to the Fzd4 CRD and the heparin mimic SOS (*Chang et al., 2015*), it is evident that the binding sites for Tspan12 and SOS on Norrin overlap (*Figure 3A*). Indeed, the Norrin-Tspan12 interaction is inhibited by increasing concentrations of SOS with a $K_i$ of 34±4 µM (*Figure 3B* and *Figure 3—figure supplement 1A*). Of note, both residues mutated in the Site 1 Norrin mutant R107E/R115E, which impair Tspan12 binding (*Figure 2E*), are also part of the SOS binding site (*Chang et al., 2015*). The deleterious effect of the Norrin mutation R115L seen in FEVR patients has previously been attributed to deficient HSPG binding; our data suggest that this mutation may also impair Tspan12 binding.

In the same structural model, the Tspan12 binding site on Norrin is adjacent to the binding site for the Fzd4 CRD on the same protomer (*Figure 3C*). While simultaneous binding of Tspan12 and Fzd4 to one Norrin protomer appears to be sterically compatible in this model, the adjacent regions of Tspan12 and Fzd4 are acidic and may compete for the same residues K102 and/or K104 on Norrin. In line with this prediction, Norrin-Tspan12 binding is negatively modulated by the Fzd4 extracellular domain: in a competition binding experiment, the amount of Norrin bound to nanodisc-embedded Tspan12 decreases slightly in response to increasing concentrations of Fzd4 CRDL, reaching a new non-zero plateau (*Figure 3D* and *Figure 3—figure supplement 1B*). This suggests that although the presence of Fzd4 may effectively weaken Norrin-Tspan12 affinity, Tspan12 and Fzd4 can simultaneously bind a single Norrin protomer. To test this directly, sequential binding of components was monitored by BLI. Biosensors loaded with nanodisc-embedded Fzd4 were bound to Norrin, followed by Tspan12 LEL, which bound in a concentration-dependent manner (*Figure 3E*). However, as Norrin is a disulfide-linked dimer, this experiment was unable to show whether Fzd4 and Tspan12 were binding to the same protomer of Norrin, because the Norrin dimer may bridge this interaction. We next immobilized Norrin on the biosensors, and bound Fzd4 CRDL at a concentration expected to saturate both sites on the Norrin dimer (5 µM; see *Figure 3—figure supplement 1C*), followed by the Tspan12 LEL (*Figure 3F*). The Tspan12 LEL-bound CRDL-saturated Norrin with a twofold weaker apparent affinity than Norrin alone. This is again consistent with a model in which Tspan12 and Fzd4 can simultaneously bind each protomer of Norrin to form a 2:2:2 complex, but with negative cooperativity.

## Tspan12 provides additional binding sites to enhance Norrin capture

In the experiments above, the receptor elements were given a degree of freedom that they do not normally possess when the full-length receptors are embedded in the plasma membrane, and any potential allosteric effects mediated by the transmembrane domain of Tspan12 could not be assessed. Fzd4 and Tspan12 appear to co-localize even in the absence of Norrin (*Ke et al., 2013*), thus the two receptors may together form a single, composite binding site with altered Norrin binding affinity compared to Fzd4 alone. We therefore investigated whether interactions within a membrane-embedded receptor heterodimer could impact Norrin binding. We were unable to co-purify Fzd4 and Tspan12 when the two proteins were co-expressed in cells, and separately- purified proteins could not be co-reconstituted into nanodiscs with appreciable efficiency. Therefore, we enforced a 1:1 Tspan12:Fzd4 dimer by co-expressing receptor constructs that were C-terminally tagged with complementary fragments of split GFP, which assembles to stably tether the receptors throughout expression,

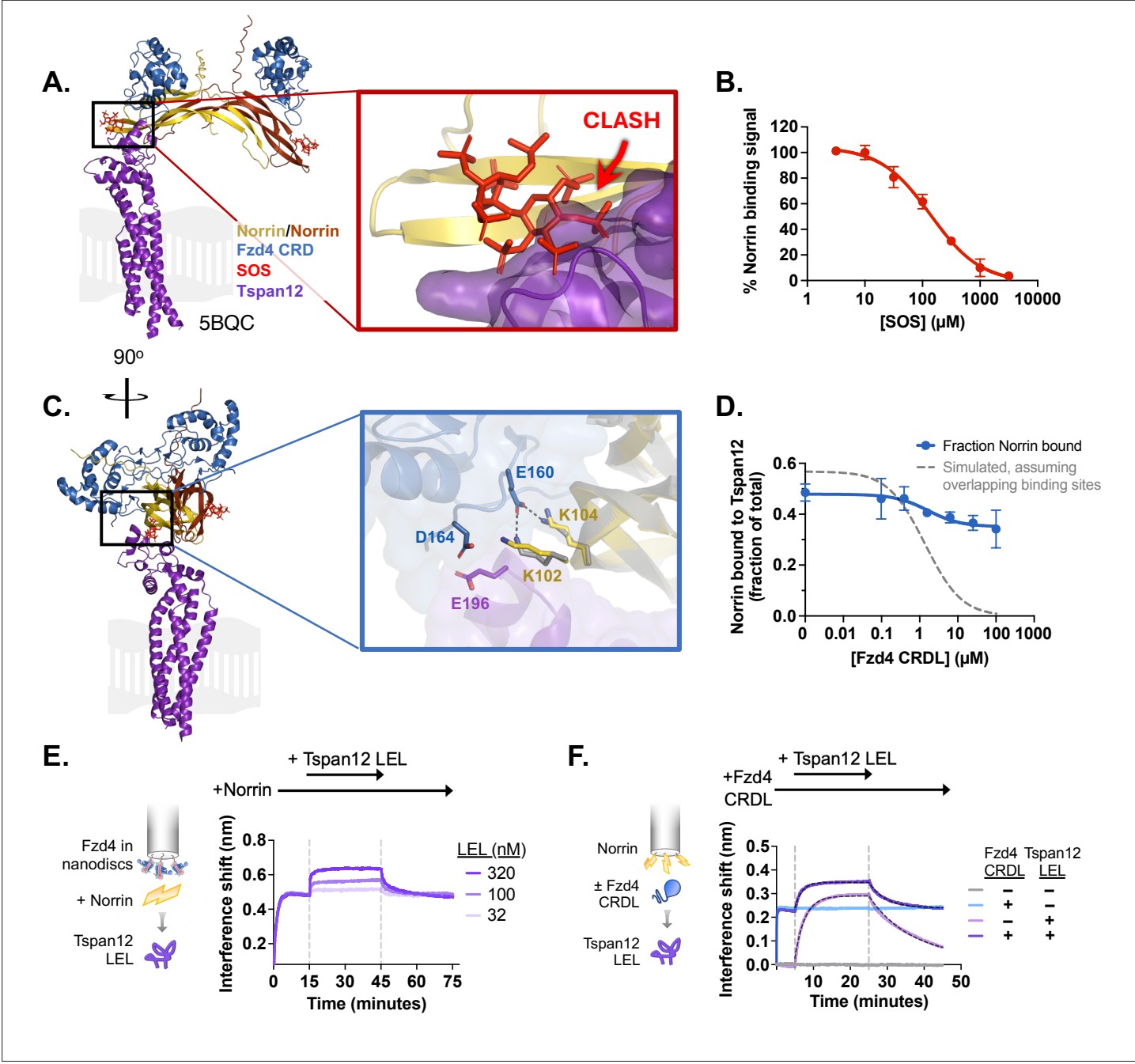

**Figure 3.** Norrin-Tspan12 binding is competitive with heparan sulfate proteoglycans but compatible with Fzd4 binding. (**A**) The AlphaFold-predicted structure of Tspan12 (purple) bound to a Norrin dimer (yellow/orange), aligned to the crystal structure (5BQC) of a Norrin dimer bound to the Fzd4 CRD (blue) and sucrose octasulfate (SOS) (red). A zoomed view of the indicated region shows an overlap in the predicted binding site of Tspan12 with that of SOS, suggesting that Tspan12 and SOS cannot bind simultaneously to Norrin. (**B**) Displacement of Norrin (32 nM) from immobilized Tspan12 by increasing concentrations of SOS, as measured by biolayer interferometry (BLI) (see *Figure 3—figure supplement 1A*). The equilibrium binding signal is plotted as a percent of signal in the absence of SOS (mean ± SD of three independent experiments), yielding a $K_i$ of 34±4 μM. (**C**) Side view of structures in A. A zoomed view of the indicated region shows that Tspan12 is predicted to occupy a site on Norrin adjacent to, but not overlapping with, the Fzd4 binding site; adjacent residues are shown. Norrin from the Tspan12-bound AlphaFold model (yellow) and the Fzd4 CRD-bound crystal structure (5BQC, gray) are overlaid. (**D**) Fzd4 CRDL does not fully compete with Tspan12-Norrin binding, as shown by equilibrium binding of 32 nM Norrin to Tspan12 immobilized on paramagnetic particles in the presence of increasing concentrations of purified Fzd4 CRDL. Bound Norrin and Norrin in the supernatant were both quantified by western blot (anti-Rho1D4; see *Figure 3—figure supplement 1B*) and used to calculate bound Norrin as a percentage of total Norrin. The expected competition curve, assuming fully competitive binding sites, was simulated (gray dashed line) given starting concentrations of 50 nM Tspan12 and 32 nM Norrin, and binding affinities of 10.4 nM for Tspan12-Norrin and 200 nM for Fzd4 CRDL-Norrin. However,

*Figure 3 continued on next page*

*Figure 3 continued*

the data better fit a model in which CRDL binding to Norrin shifts Norrin affinity for Tspan12 (blue line). Data represent mean ± SD of three replicates. (**E**) BLI traces of a ternary Fzd4-Norrin-Tspan12 large extracellular loop (LEL) complex. Biosensors loaded with nanodisc-embedded Fzd4 were first saturated with 100 nM Norrin, then bound to 32, 100, or 320 nM Tspan12 LEL. (**F**) BLI traces of ternary complex formation. Biosensors loaded with maltose binding protein (MBP)-tagged Norrin were pre-incubated in buffer or saturated with Fzd4 CRDL (5 µM), then bound to 100 nM MBP-tagged Tspan12 LEL (±5 µM CRDL). Tspan12 LEL did bind to Norrin in the presence of the Fzd4 CRDL (dark purple; apparent $K_D$ = 27 ± 2.8 nM), albeit more weakly than it bound to Norrin alone (light purple; apparent $K_D$ = 16 ± 1.8 nM; see also *Figure 1E*). Binding affinities were obtained from kinetic fits (black dotted line) to association and dissociation traces of MBP-LEL (100 nM) from three independent experiments.

The online version of this article includes the following source data and figure supplement(s) for figure 3:

**Source data 1.** Interference shift and band quantification values used to generate *Figure 3B and D*.

**Figure supplement 1.** Tspan12 Norrin binding can be competed with sucrose octasulfate (SOS) but not the Fzd4 CRDL domain.

**Figure supplement 1—source data 1.** Original files of western blots in *Figure 3—figure supplement 1*.

**Figure supplement 1—source data 2.** Labeled western blots in *Figure 3—figure supplement 1*.

purification, and insertion into nanodiscs (*Figure 4—figure supplement 1*; *Bruguera et al., 2022*). We assessed the stoichiometry of the heterodimer preparations by quantitative western blotting (*Figure 4—figure supplement 2*), confirming that these preparations contain 1.17±0.11 Tspan12 and 1.11±0.12 Fzd4 receptors per nanodisc (i.e., per two copies of MSP belt protein), compared to 1.17±0.05 Tspan12 receptors per nanodisc in the absence of Fzd4. We have shown previously that our preparations of Fzd4 are also predominantly monomeric in nanodiscs (*Bruguera et al., 2022*; *Mahoney et al., 2022*).

Norrin binds to the Tspan12/Fzd4 dimer with a $K_D$ of 2.18±0.10 nM, which is slightly tighter than Norrin's affinity for Fzd4 alone (3.29±0.17 nM), and tighter than Norrin's affinity for Tspan12 alone (10.4±1.2 nM) (*Figure 4A*, *Figure 4—figure supplement 3A–C*, and *Supplementary file 1*). Interestingly, Fzd4 and Tspan12/Fzd4 differ most in their ability to bind Norrin at lower concentrations (≤1 nM), yielding Hill slopes that differ significantly and reproducibly for the different receptor assemblies (1.7±0.1 for the Tspan12/Fzd4 dimer vs. 2.8±0.6 for Fzd4 alone, and 1.0±0.1 for Tspan12 alone).

Given this slight increase in affinity, we wondered whether Tspan12 and Fzd together can form a composite, higher affinity binding site for each Norrin protomer, or whether the heterodimer simply displayed increased avidity and was able to simultaneously bind two disulfide-linked Norrin protomers. To investigate this, we purified a monomeric mutant of Norrin, C93A/C95A/C131A, which lacks the three cysteines that link the dimer together (*Ke et al., 2013*). This mutant elutes slightly later than WT Norrin on size exclusion chromatography, runs as a monomer on non-reducing SDS-PAGE, and produces particles of a size consistent with a monomer by negative stain electron microscopy at 100 nM (*Figure 4—figure supplement 4*). If heterodimeric Tspan12 and Fzd4 together form an extended, composite binding site for a single Norrin protomer, then we would expect its affinity for monomeric Norrin to be greater than that of Fzd4 alone. Purified monomeric Norrin binds Fzd4 and the Tspan12/Fzd4 heterodimer with similar affinities (13.2±1.2 nM [Hill slope = 2.0 ± 0.3] for Fzd4, vs. 11.5±1.5 nM [Hill slope = 1.0 ± 0.1] for Tspan12/Fzd4) (*Figure 4B* and *Figure 4—figure supplement 3D–F*). This suggests that Tspan12 and Fzd4 do not cooperate to form a single, higher affinity binding site for each Norrin protomer. Monomeric Norrin bound Tspan12 alone much more weakly and did not reach saturation at 1 µM (*Figure 4B*). This behavior is consistent with the structural prediction that the binding site for Tspan12 on Norrin spans across part of the dimer interface (*Figure 2C and D*). However, it is also possible that Norrin dimerization is required for the structural stability of the β strands that form the Tspan12 binding site.

If Tspan12 does not directly increase the affinity of Fzd4 for Norrin, we next hypothesized that it could be helping cells capture Norrin upstream of signaling. We transfected Expi293 cells with Fzd4, Tspan12, or both receptors together, incubated them with purified Norrin, and quantified cell-surface Norrin binding by flow cytometry (*Figure 4C*). Fzd4-transfected cells capture more Norrin when Tspan12 is co-transfected, even though Tspan12 co-expression slightly decreases Fzd4 surface expression (*Figure 4—figure supplement 3G*). This is true at all concentrations of Norrin tested, although the effect is greatest at low concentrations of Norrin (≤1 nM), mirroring the results of binding assays with purified protein.

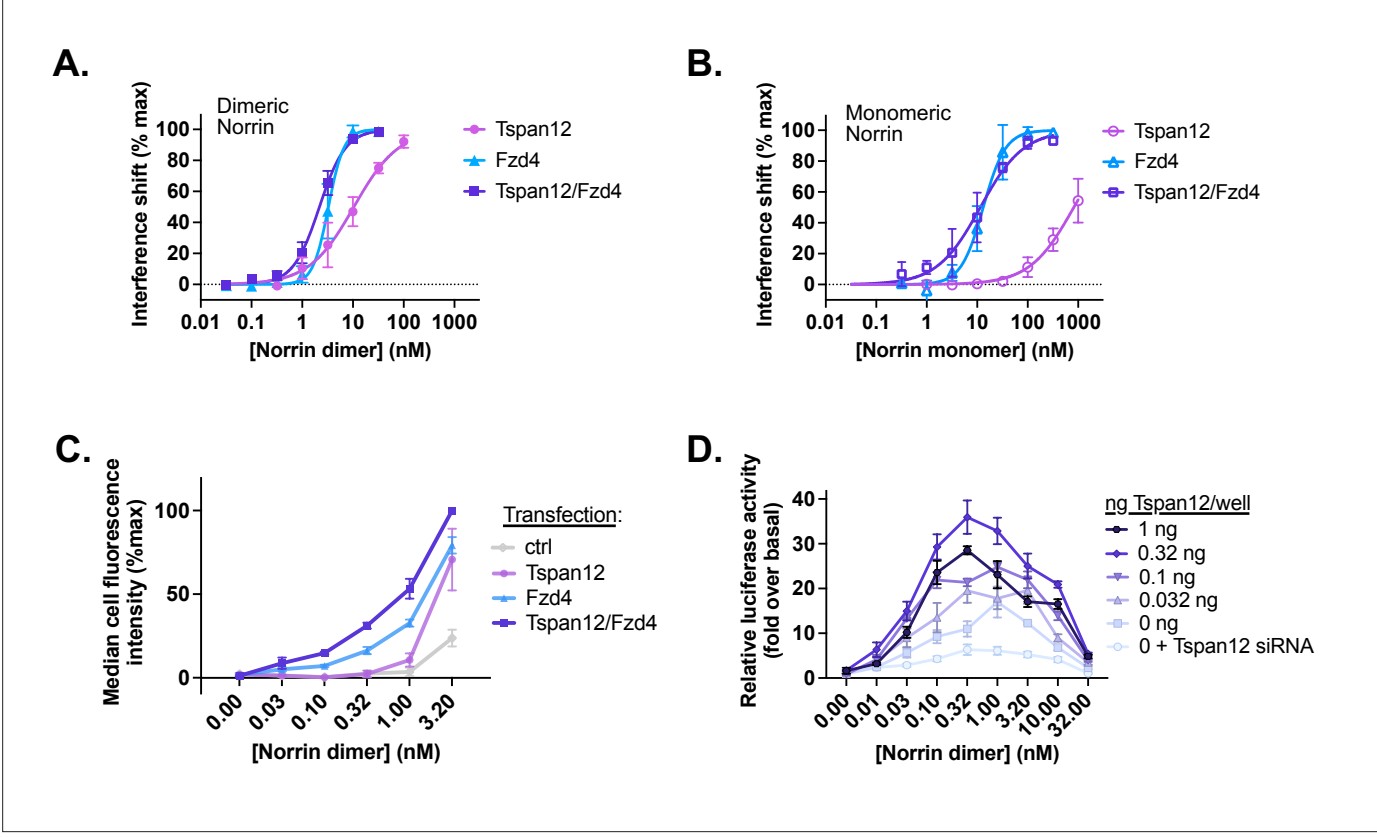

**Figure 4.** Tspan12 enhances Norrin-Fzd4 binding, cell-surface binding, and Norrin-stimulated β-catenin signaling at low Norrin concentrations. (**A**) Steady-state binding curves of monomeric Tspan12ΔC, monomeric Fzd4, or heterodimeric Tspan12ΔC/Fzd4ΔC receptors in biotinylated nanodiscs binding to dimeric or (**B**) monomeric (C93A/C95A/C131A) Norrin by biolayer interferometry (BLI). Steady-state binding signal is plotted as a percent of $B_{max}$ for three independent replicates (mean ± SD). Affinities and kinetic constants are reported in ***Supplementary file 1***. (**C**) Indicated concentrations of Norrin-1D4 dimer binding to Expi293 cells transfected with Fzd4, Tspan12, or both Fzd4 and Tspan12, detected with fluorescently labeled Rho1D4 antibody and quantified by flow cytometry. Mean ± SD of three independent experiments are plotted. Co-transfection of Tspan12 increased Norrin recruitment to Fzd4-transfected cells at 0.1, 0.32, 1, and 3.2 nM Norrin (two-tailed t-test p-values of 0.00026, 0.00079, 0.0049, and 0.0018, respectively). (**D**) β-Catenin pathway activation resulting from increasing concentrations of Norrin was assessed in Fzd1/2/4/5/7/8-knockout HEK293T cells transfected with Tspan12 siRNA or increasing amounts of Tspan12 plasmid, along with Fzd4 and TopFlash luciferase reporter plasmids. Data are plotted as mean ± SD from triplicate wells are representative of three independent experiments.

The online version of this article includes the following source data and figure supplement(s) for figure 4:

**Source data 1.** Interference shift, cell fluorescence, and luciferase activity values used to generate ***Figure 4A–D***.

**Figure supplement 1.** Purification of Fzd4/Tspan12 dimer and insertion into nanodiscs.

**Figure supplement 1—source data 1.** Original files of gels in ***Figure 4—figure supplement 1***.

**Figure supplement 1—source data 2.** Labeled gels in ***Figure 4—figure supplement 1***.

**Figure supplement 2.** Stoichiometry of receptors in nanodiscs was determined by quantitative western blot.

**Figure supplement 2—source data 1.** Original files of western blots in ***Figure 4—figure supplement 2***.

**Figure supplement 2—source data 2.** Labeled western blots in ***Figure 4—figure supplement 2***.

**Figure supplement 3.** Tspan12 enhances Norrin recruitment.

**Figure supplement 4.** Purification and validation of monomeric Norrin.

**Figure supplement 4—source data 1.** Original file of gel in ***Figure 4—figure supplement 4***.

**Figure supplement 4—source data 2.** Labeled gel in ***Figure 4—figure supplement 4***.

If the role of Tspan12 is to help cells capture Norrin, and if its impact on Norrin capture is most evident at low concentrations of Norrin, we would expect it to have the largest effect on β-catenin signaling at low concentrations of Norrin – i.e., Norrin should more potently stimulate β-catenin signaling in cells expressing Tspan12. To assess signaling, we transfected Fzd1/2/4/5/7/8-knockout

HEK293T cells with Tspan12 siRNA or increasing amounts of Tspan12 plasmid alongside fixed amounts of Fzd4 and a β-catenin-responsive reporter plasmid (TopFlash). We then measured luciferase activity in response to increasing amounts of recombinant Norrin (*Figure 4D*). Norrin stimulation results in a bell-shaped dose-response curve, consistent with its role in heterodimerizing Fzd4 and LRP5/6 (i.e., excessive Norrin concentrations will fully saturate receptor binding sites, inhibiting receptor dimerization). In all conditions, Norrin exhibited maximal activity at or below 1 nM, the same concentration regime in which we see the largest effect of Tspan12 on Norrin binding to Fzd4-containing nanodiscs (*Figure 4A*) and Fzd4-expressing cells (*Figure 4C*). Contrary to our expectation, in the cell-based signaling assay, Tspan12 does not appear to have a marked effect on Norrin potency, and instead increases the amplitude of response to Norrin at all concentrations of Norrin tested.

## Tspan12 does not directly enhance formation of the Norrin/Fzd4/LRP/Dvl signaling complex

Because β-catenin signaling depends on the ability of ligands to bring together Fzd and LRP5/6, we next hypothesized that Tspan12 might increase Norrin efficacy by forming a complex with Norrin, Fzd4, and LRP5/6 to exert a direct effect on downstream signaling (*Figure 5A*). Tspan12 has been shown to co-localize and co-internalize with Fzd4, LRP5, and Norrin in Norrin-stimulated cells (*Zhang et al., 2017*). As part of this complex, Tspan12 might modulate the direct Fzd4-Dvl interaction on the intracellular side of the membrane, since tetraspanins have been known to recruit intracellular partners (*Lapalombella et al., 2012*; *Zhang et al., 2001*). This hypothesis mirrors a proposed role of Gpr124 in zebrafish, which co-IPs with Dvl (*Eubelen et al., 2018*). An increase in Fzd4 affinity for the Dvl2 DEP domain in the presence of Tspan12, either with or without Norrin, would explain the Tspan12-mediated enhancement of signaling. To test this directly, we inserted Fzd4 alone or Tspan12/Fzd4 heterodimers into nanodiscs containing 5% PI(4,5)P$_2$, which enhances DEP recruitment (*Mahoney et al., 2022*), and measured DEP binding by BLI (*Figure 5B*). Contrary to our hypothesis, we found that the affinity of the Dvl2 DEP domain for the Tspan12/Fzd4 heterodimer is not significantly different than its affinity for Fzd4 alone (*Figure 5C*) and is unchanged in the presence of Norrin (*Figure 5D*). This is consistent with our previous findings that neither Wnt ligands nor LRP6 allosterically modulate Fzd-DEP binding (*Mahoney et al., 2022*).

Finally, we wondered whether Tspan12 might cooperatively enhance Norrin-LRP5/6 binding to increase Norrin signaling efficiency. Of note, while LRP6 is known to use the β-propeller-EGF repeats 1 and 2 (E1E2) within its extracellular domain to bind Norrin (*Ke et al., 2013*), no structural information on Norrin-LRP5/6 binding is available, and AlphaFold was unable to predict the structure of a Norrin-LRP5/6 complex with high confidence (data not shown). It has been proposed that Norrin uses a positively charged patch composed of residues K54, R90, R97, G112, and R121 to bind LRP6 *Chang et al., 2015*; of these residues, R121 forms part of the AlphaFold-predicted binding site for Tspan12 (*Figure 2B*), which predicts that Tspan12 and LRP5/6 may compete for the same binding site on Norrin. Indeed, we found that Norrin can be completely displaced from nanodisc-embedded Tspan12 by increasing amounts of purified LRP6 E1E2 domain (*Figure 5E* and *Figure 5—figure supplement 1A*). We calculated the K$_i$ to be 1.06 μM, which agrees with previous LRP6-Norrin affinity measurements (*Chang et al., 2015*; *Ke et al., 2013*). In keeping with this result, the Tspan12 LEL displayed concentration-dependent inhibition of Norrin binding to a larger portion of LRP6 (including the full extracellular and transmembrane domains but lacking a portion of the C-terminus) (*Figure 5—figure supplement 1B and C*). While LRP5, but not LRP6, has been genetically implicated in Norrin-directed retinal vascularization, we expect the binding site for Norrin to be conserved in both LRP5 and LRP6 because LRP6 binds Norrin, transduces Norrin-stimulated and Tspan12-enhanced TOPFLASH signaling, and is highly homologous to LRP5 (*Chang et al., 2015*; *Ke et al., 2013*; *Zhou and Nathans, 2014*).

These experiments imply that LRP5/6 and Tspan12 compete for the same binding site on Norrin or that their simultaneous binding is otherwise sterically incompatible. Furthermore, it suggests that Tspan12 does not incorporate into the Norrin-Fzd4-LRP5/6 signaling complex via interactions with Norrin. If Tspan12 competes with LRP5/6 for Norrin binding, it should inhibit signaling, which we found to be the case in cells. In TopFlash signaling assays, we observed a bell-shaped curve in response to transfected Tspan12, where high levels of Tspan12 completely inhibit Norrin-stimulated signaling (*Figure 5F*). Wnt3a signaling was moderately inhibited at the highest level of transfected

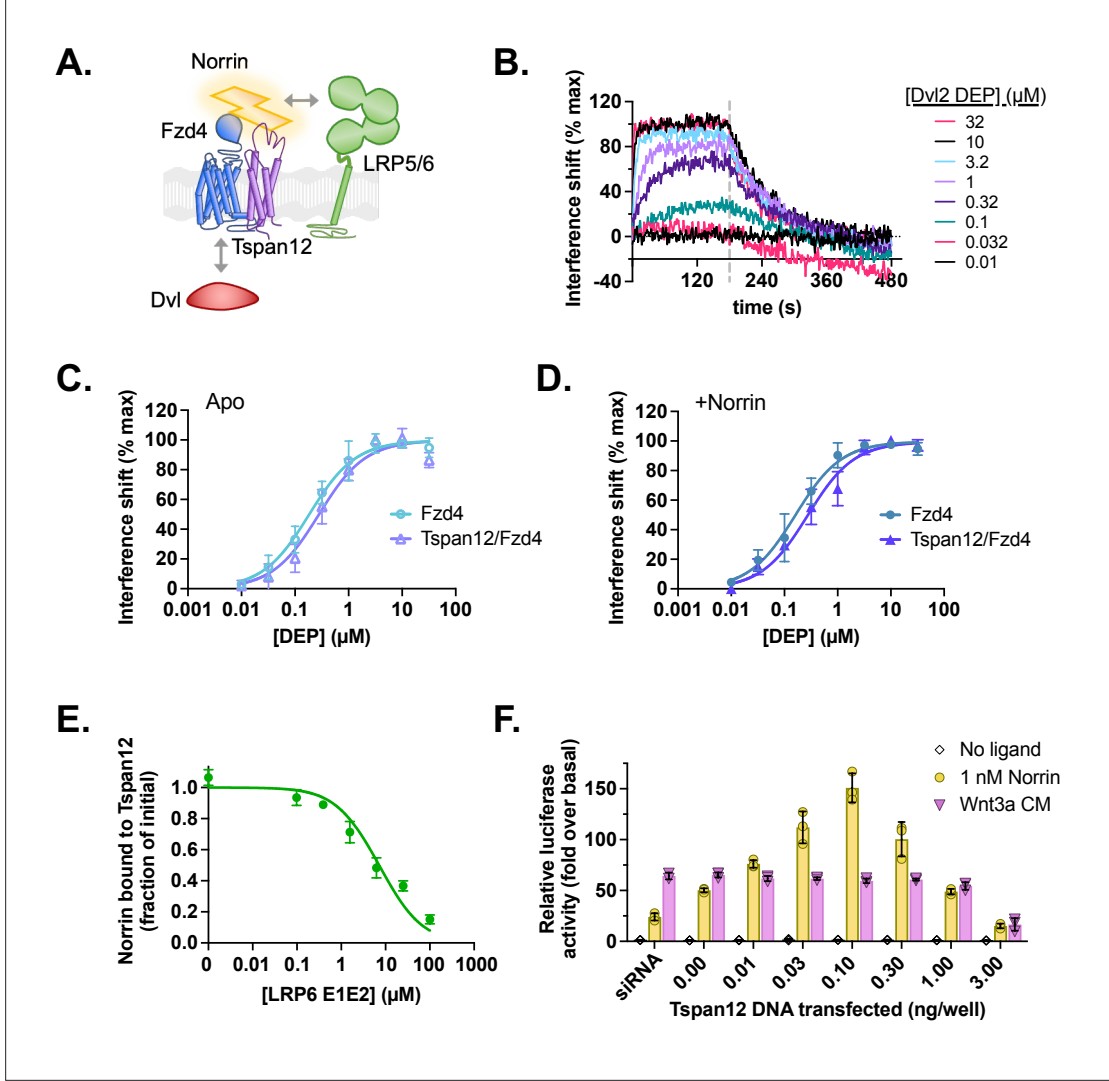

**Figure 5.** Tspan12 does not directly enhance formation of a Norrin-LRP5/6-Fzd4-Dvl signaling complex. (**A**) Hypothesis: Tspan12 could enhance Norrin signaling by enhancing interactions within the Norrin-LRP5/6-Fzd4-Dvl complex, including Fzd-Dvl binding and Norrin-LRP binding. (**B**) Representative biolayer interferometry (BLI) traces of the Dvl2 DEP domain associating to and dissociating from Fzd4 in nanodiscs containing 75:20:5 POPC:Ccholesterol:PIP$_2$. (**C**) Equilibrium binding of the Dvl2 DEP domain to Fzd4 monomer or Tspan12/Fzd4 heterodimer in nanodiscs; affinities ± SEM are 183±24 and 279±46 nM, respectively. (**D**) Equilibrium binding of the Dvl2 DEP domain to Fzd4 monomer or Tspan12/Fzd4 heterodimer nanodiscs, each pre-saturated with 10 nM Norrin. Binding affinities are 161±21 and 274±39 nM (mean ± SEM), respectively, determined from three independent replicates. Affinities and kinetic constants are reported in *Supplementary file 1*. (**E**) The LRP6 E1E2 domain fully competes with Tspan12-Norrin binding, as shown by decreased equilibrium binding of 32 nM Norrin to Tspan12 immobilized on paramagnetic particles in the presence of increasing concentrations of purified LRP6 E1E2 domain. Norrin was quantified by western blot (anti-Rho1D4; see *Figure 5—figure supplement 1*) and plotted as a percent of bound Norrin in the absence of LRP6 E1E2. The curve was fit to a competitive binding model using known binding affinities of 10.4 nM for Tspan12-Norrin and starting concentrations of 50 nM Tspan12 and 32 nM Norrin; the best fit reported a Norrin-LRP6 E1E2 binding affinity of 1.06 µM (95% CI 0.747–1.51 µM). Data represent mean ± SD of three replicates. (**F**) β-Catenin transcriptional activity in response to no ligand, 1 nM recombinant Norrin, or Wnt3a conditioned media (Wnt3a CM) in Fzd1/2/4/5/7/8-knockout HEK293T cells transfected with Tspan12 siRNA or indicated amount of Tspan12_pTT5 plasmid, along with Fzd4 and TopFlash luciferase reporter plasmids. Data are plotted as mean ± SD from n=3 replicate wells.

The online version of this article includes the following source data and figure supplement(s) for figure 5:

**Source data 1.** Interference shift, band quantification, and luciferase activity values used to generate *Figure 5C–F*.

**Figure supplement 1.** Tspan12 and LRP6 E1E2 compete for Norrin binding.

**Figure supplement 1—source data 1.** Original file of western blot in *Figure 5—figure supplement 1*.

**Figure supplement 1—source data 2.** Labeled western blot in *Figure 5—figure supplement 1*.

Tspan12, an effect that may be due to a negative influence of Tspan12 transfection on cell-surface expression of Fzd4 (*Figure 4—figure supplement 3G*). However, Norrin signaling was inhibited to a greater extent and at lower levels of transfected Tspan12, which is consistent with a model in which Tspan12 competes with LRP6 for Norrin binding.

## Discussion

The spatiotemporal specificity of Wnt- and Norrin/β-catenin signaling is tightly controlled by a growing number of known modulators. In this study, we used biochemical approaches to investigate the molecular mechanism by which the Norrin-specific co-receptor Tspan12 enhances Norrin/β-catenin signaling. We have demonstrated direct, high-affinity binding of Norrin to Tspan12, and our mutagenesis studies corroborate a binding site on the Tspan12 LEL predicted by AlphaFold Multimer. Several mutations in the predicted interface have been associated with Norrie disease, FEVR, and other diseases of the retinal vasculature. The missense mutations of clinical significance in the Tspan12 LEL are severe enough that they would be expected to disrupt the LEL fold or lead to Tspan12 aggregation (e.g. mutations to/from cysteine or proline). The Norrie disease-associated mutation T119P in Norrin would also be expected to disrupt Norrin folding, but other clinically relevant Norrin mutations at positions R115 and R121 likely exert their influence by disrupting the Norrin-Tspan12 interface. Using mechanistic models of other cell-surface modulators of Wnt/β-catenin signaling as a starting point, we probed how the direct Tspan12-Norrin interaction might promote Norrin/β-catenin signaling.

### Tspan12 captures Norrin upstream of signaling

Tspan12 has been proposed to enhance Norrin-Fzd4 affinity, supported initially by findings that Tspan12 rescues Norrin cell-surface binding and Norrin-stimulated signaling in cells when mutations disrupt the Norrin-Fzd4 interface (*Lai et al., 2017*). We now have shown that Tspan12 also enhances binding of Norrin to Fzd4-containing nanodiscs and Fzd4-expressing cells. This effect is most evident at very low concentrations of Norrin, which could explain why previous experiments using Norrin-conditioned media did not show Tspan12-mediated differences in cell-surface Norrin binding (*Junge et al., 2009*). Tspan12 might facilitate the Norrin-Fzd4 interaction by (1) increasing the number of cell-surface binding sites for Norrin and thereby increasing the local concentration of Norrin, similar to a proposed role for HSPGs in signaling through Wnt (*Baeg et al., 2001*; *Reichsman et al., 1996*) or (2) serving as a cofactor that forms a complex with Fzd4 and Norrin to increase Norrin-Fzd4 affinity, similar to a proposed role of heparin, which increases Norrin capture by Fzd4 in ELISA (*Smallwood et al., 2007*). On the basis that the Fzd4 CRDL and Tspan12 bind Norrin with negative cooperativity, we propose that Tspan12 does not directly enhance Norrin-Fzd4 affinity. Rather, it enhances Norrin capture and increases local Norrin concentration. As Tspan12 co-localizes with Fzd4 on the cell surface (*Junge et al., 2009*; *Ke et al., 2013*; *Lai et al., 2017*), Tspan12 can hand off captured Norrin to nearby Fzd4 for signaling, a process facilitated by the negative cooperativity of Norrin-Fzd4 and Norrin-Tspan12 binding.

In addition to co-localizing with Fzd4, Tspan12 has been shown to co-localize with the Fzd4-Norrin-LRP5 signaling complex at the cell surface and in endosomes (*Zhang et al., 2017*). Yet, we found that Tspan12 competes with LRP6 for Norrin binding, suggesting that Tspan12 does not remain bound to Norrin in a quaternary complex with Fzd4 and LRP5/6 co-receptors. These results do not preclude the possibility that Tspan12 might incorporate into the signalosome by interacting with Fzd4 even after passing off Norrin. Alternatively, Tspan12 and LRP5/6 may each bind one Norrin protomer and thus be bridged as a complex through Norrin dimerization, averting the need for complete dissociation of Tspan12, a potentially inefficient process since Norrin binds Tspan12 with a relatively high affinity even in the presence of Fzd4.

Within the signalosome, Tspan12 could enhance Norrin-stimulated signaling by promoting downstream interactions like Dvl recruitment by Fzd. However, we found that Tspan12 did not affect Fzd4 affinity for the Dvl2 DEP domain in the absence of Norrin, nor did Tspan12 allow Norrin to allosterically enhance Fzd4-DEP affinity. These observations suggest that Tspan12 exerts its influence on Norrin/β-catenin signaling primarily by enhancing extracellular ligand capture, but we cannot rule out the possibility that Tspan12 may aid in Dvl recruitment by binding other regions of Dvl. An alternative, untested

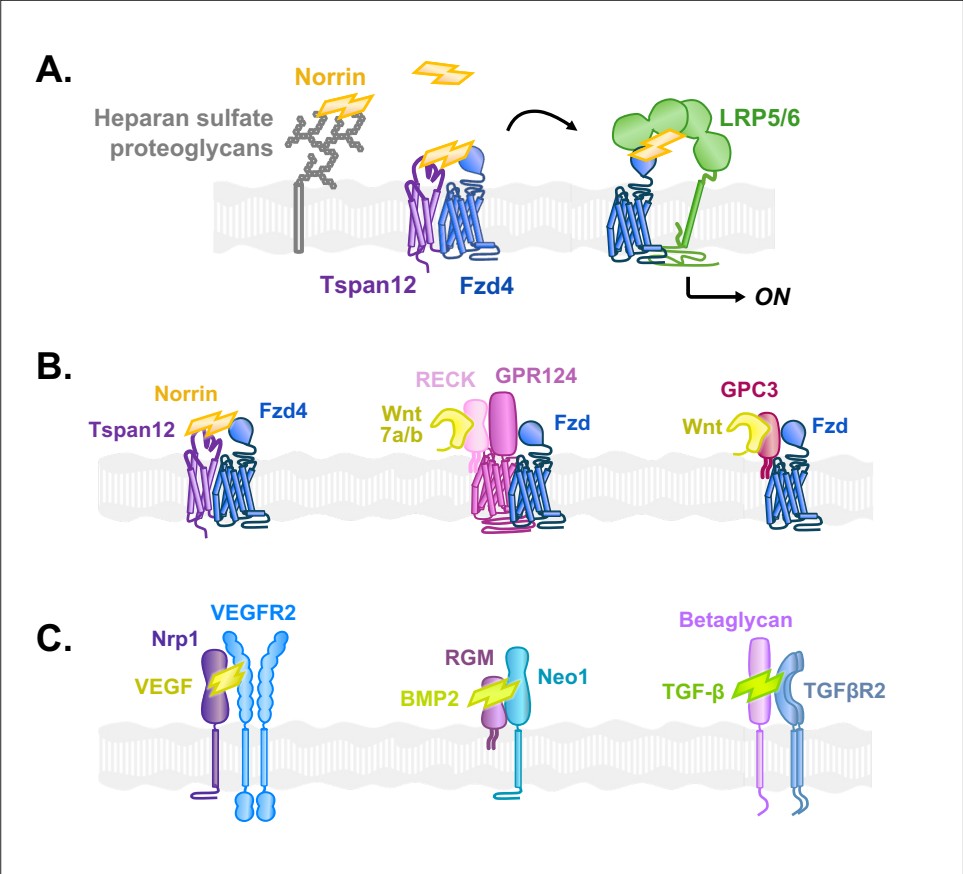

**Figure 6.** Diverse co-receptors facilitate growth factor signaling by capturing and delivering ligands to their target receptors. (**A**) Model: Norrin is captured by Tspan12 or heparan sulfate proteoglycans (HSPGs) and is handed off to Fzd4 for association with LRP5/6 and subsequent signaling. Norrin binding to cell-surface Fzd4 is enhanced when HSPGs concentrate Norrin at the cell surface. In contrast, Tspan12 directly and specifically delivers Norrin to co-localized Fzd4. (**B**) In the β-catenin signaling pathway, *Left:* Tspan12 captures Norrin and co-localizes with Fzd4, delivering Norrin to Fzd4. *Middle:* Likewise, RECK binds Wnt7a/b and co-localizes with Fzd via GPR124, delivering Wnt7a/b to Fzd. *Right:* Glypican-3 (GPC3) also binds both Fzd and Wnt to deliver Wnt to Fzd and enhance signaling. (**C**) Structurally diverse co-receptors play a similar role to Tspan12 in various pathways activated by cystine knot growth factors. *Left:* Neuropilin-1 (Nrp1) captures vascular endothelial growth factor A (VEGF) and co-localizes with the VEGF receptor 2 (VEGFR2) to specifically deliver VEGF to VEGFR2. *Middle:* The repulsive guidance molecule (RGM) binds bone morphogenic protein 2 (BMP2) as well as neogenin-1 (Neo1) to facilitate BMP signaling. *Right:* Betaglycan captures transforming growth factor β1 (TGF-β) and presents it to TGF-β receptor type 2 (TGFβR2).

mechanism may involve Tspan12-mediated recruitment or activation of phosphoinositide kinases such as PI4KIIα, which binds other tetraspanins (*Berditchevski et al., 1997*; *Yauch et al., 1998*; *Yauch and Hemler, 2000*) and is required for Wnt-stimulated PIP$_2$ generation that enhances Dvl recruitment to Fzd and subsequent LRP5/6 phosphorylation (*Mahoney et al., 2022*; *Pan et al., 2008*).

Overall, our results are consistent with a model in which Tspan12 captures Norrin and hands it off like a baton to Fzd4 before formation of the Fzd4-Norrin-LRP5/6 complex (*Figure 6A*). Thus, the observed effects of Tspan12 on Norrin/β-catenin signaling are driven by a Tspan12-dependent increase in the local concentration of Norrin, which provides avidity to enhance the efficiency of Fzd4-Norrin-LRP5/6 complex formation at low Norrin concentrations.

Such a mechanism may represent a more general theme among modulators that enhance Wnt/β-catenin signaling (*Figure 6B*). The co-receptor Gpr124 similarly co-localizes with Fzd (*America et al., 2022*; *Eubelen et al., 2018*) and captures Wnt7a/7b through RECK, thereby delivering Wnt to Fzd. The HSPG Glypican-3 also binds both Fzd and Wnt to enhance signaling (*Capurro et al., 2014*; *Li et al., 2019*). Interestingly, these systems rely on both Wnt and Fzd binding to function: RECK without

Gpr124 captures Wnt7a/b but does not deliver it to Fzd and does not mediate signaling enhancement on its own (*America et al., 2022*; *Cho et al., 2019*; *Eubelen et al., 2018*; *Vallon et al., 2018*). Likewise, Glypican-6, which can bind Wnt but not Fzd, does not promote Wnt signaling (*Capurro et al., 2014*). Similarly, Tspan12 likely enhances signaling through its ability to both bind Norrin and co-localize with Fzd4 on the cell surface, thus promoting the formation of a Norrin-Fzd4 complex. However, the question of how Tspan12 co-localizes with Fzd4 remains to be addressed.

Outside of Wnt signaling, various other growth factors are controlled by the same mechanism: co-receptors for vascular endothelial growth factor A, bone morphogenic protein, and transforming growth factor β bridge these ligands and their receptors to enhance signaling (*Healey et al., 2015*; *Hinck et al., 2016*; *López-Casillas et al., 1993*; *Soker et al., 2002*; *Figure 6C*). Interestingly, these ligands share the same cystine knot fold as Norrin, and similarly dimerize or oligomerize their cognate receptors to initiate signaling.

It has previously been proposed that Tspan12 directly enhances Fzd4-Fzd4 (*Junge et al., 2009*) or Fzd4-LRP5 (*Knoblich et al., 2014*) interactions. These models were supported by co-IP data and bolstered by observations that tetraspanins cluster in membrane nanodomains with their binding partners (*Zuidscherwoude et al., 2015*). According to such a model, Tspan12-mediated clustering or localization of Fzd4 would be expected to enhance not only Norrin- but also Wnt-mediated signaling, which it does not (*Junge et al., 2009*; *Lai et al., 2017*; *Zhang et al., 2017*; *Figure 5F*). This can be reconciled if Tspan12-mediated receptor clustering is Norrin-dependent: e.g., a Tspan12-rich membrane domain may capture Norrin to establish a high local Norrin concentration and consequently nucleate or stabilize clusters of Norrin-Fzd4 complexes and enable Fzd4-LRP5/6 crosslinking.

Tspan12 and Fzd4 exist as ligand-independent homodimers, heterodimers, and possibly larger oligomeric clusters in the membrane (*Ke et al., 2013*), but due to the split GFP reconstitution strategy utilized in this study our results do not capture Tspan12's role in the context of a putative receptor multimer with higher-order stoichiometry (e.g., 2:2 Tspan12:Fzd4). Our reconstituted system also may not capture physiological receptor phosphorylation or other post-translational modifications that could be necessary for Tspan12 to exert a direct role in Fzd4-Norrin or Fzd4-Dvl binding. Additionally, the lipid composition, membrane fluidity, and membrane curvature that these receptors encounter in the cell membrane are not fully recapitulated in nanodiscs and could be important for Tspan12 function. Due to these limitations, further experiments conducted in cells will be required to examine the predictions of our model in the context of physiological receptor stoichiometry, post-translational modifications, and plasma membrane environment.

## Tspan12 expression dictates specificity of Norrin/β-catenin signaling

The sensitivity of Norrin/β-catenin signaling to Tspan12 levels (*Figure 4D*) underscores Tspan12 expression as a mechanism by which cell response to secreted Norrin can be tuned. As such, it provides a strategy to spatiotemporally regulate development. For example, in the retina, Tspan12 is expressed in vascular endothelial cells while Fzd4 and LRP5 are more broadly expressed (*Junge et al., 2009*). Consequently, Tspan12 expression dictates which cells preferentially bind and respond to Norrin, thereby directing proper angiogenesis.

Overcoming on-target toxicity in off-target tissues presents a major challenge in therapeutically targeting the Wnt/β-catenin pathway (*Diamond et al., 2020*; *Tabernero et al., 2023*), as systemically administered drugs do not benefit from the spatial specificity of endogenous ligands, whose expression varies by tissue and cell type (*Murphy et al., 2022*; *Rawal et al., 2006*; *Summerhurst et al., 2008*). Tspan12 and other co-receptors can serve as specific drug targets, as well as mechanistic inspiration, for the development of new ligand- and tissue-specific therapies. Targeting pathway modulators that have restricted expression patterns would achieve spatial specificity by limiting therapeutic activity to a select subset of tissues (*Chouinard et al., 2016*). More broadly, a mechanistic understanding of Wnt/β-catenin pathway modulators like Tspan12 could inspire the development of therapeutics to inhibit or activate the pathway in a cell- and tissue-specific manner.

# Materials and methods

## Key resources table

| Reagent type (species) or resource | Designation | Source or reference | Identifiers | Additional information |
|---|---|---|---|---|
| Recombinant DNA reagent | untagged Tspan12 | DNASU | Clone HsCD00045716 | Subcloned into pTT5 vector |
| Recombinant DNA reagent | Tspan12ΔC (human) | This paper | | pVL1393 vector; residues 1–252; Rho-1D4-tagged |
| Recombinant DNA reagent | Tspan12 (full-length, human) | This paper | RRID:Addgene_216381 | pTT5 vector; Rho-1D4 tagged; deposited at Addgene (#216381) |
| Recombinant DNA reagent | Tspan12-LEL11 (human) | PMID:28658627 | | Rho-1D4-tagged and subcloned into pVL1393 vector |
| Recombinant DNA reagent | Tspan12LEL (human) | This paper | RRID:Addgene_216383 | pAcGP67a vector; residues 118–218; MBP-tagged; Deposited at Addgene (#216383) |
| Recombinant DNA reagent | Fzd4 (mouse) | PMID:35998232 | RRID:Addgene_216378 | pFastBac1 vector; residues 42–537; FLAG-tagged; Deposited at Addgene (#216378) |
| Recombinant DNA reagent | HA-FLAG-Fzd4ΔC-3C-GFP1-10OPT-ETVV (mouse) | PMID:35074428 | RRID:Addgene_216380 | pVL1393 vector; residues 42–513; Deposited at Addgene (#216380) |
| Recombinant DNA reagent | Tspan12ΔC-His-3C-GFP11 (human) | This paper | RRID:Addgene_216382 | pVL1393 vector; residues 1–252; Deposited at Addgene (#216382) |
| Recombinant DNA reagent | LRP6 (human) | PMID:35074428 | | pEZT-BM vector; residues 20–1439; FLAG-tagged |
| Recombinant DNA reagent | LRP6 E1E2 (human) | PMID:22000856 | | pAcGP67a vector; 10xHis-tagged |
| Recombinant DNA reagent | MESD (human) | PMID:35074428 | | pEZT-BM vector |
| Recombinant DNA reagent | Norrin (human) | PMID:35074428 | RRID:Addgene_216384 | pAcGP67a vector; residues 33–133; MBP- and Rho-1D4 tags; Deposited at Addgene (#216384) |
| Recombinant DNA reagent | Monomeric Norrin | This paper | RRID:Addgene_216386 | pAcGP67a vector; residues 33–133; MBP- and Rho-1D4-tagged; C93A/C95A/C131A; Deposited at Addgene (#216386) |
| Recombinant DNA reagent | Fzd4 CRDL (mouse) | PMID:30104375 | RRID:Addgene_216379 | pAcGP67a vector; residues 42–203; 6xHis-tagged; Deposited at Addgene (#216379) |
| Recombinant DNA reagent | DEP (DEP domain from mouse Dvl2) | PMID:35998232 | RRID:Addgene_216386 | pCDF vector; residues 416–510; MBP-tagged; Deposited at Addgene (#216386) |
| Recombinant DNA reagent | GFP nanobody | PMID:20945358 | | pET26(+) vector; with pelB signal sequence and 6xHis tag |
| Recombinant DNA reagent | MSP1D1 | Addgene | RRID:Addgene_20061 | Sligar lab |
| Recombinant DNA reagent | MSP1E3D1 | Addgene | RRID:Addgene_20066 | Sligar lab |
| Commercial assay or kit | EZ-link NHS-PEG4-Biotin | Thermo | Cat. #: A39259 | |
| Other | 16:0–18:1 phosphatidyl choline (POPC) | Avanti Polar Lipids | Cat. #: 850457 | For nanodisc reconstitution (see Materials and methods section) |
| Other | 16:0–18:1 phosphatidylglycerol (POPG) | Avanti Polar Lipids | Cat. #: 840457 | For nanodisc reconstitution (see Materials and methods section) |
| Other | 18:0–20:4 brain phosphatidylinositol-4,5-bisphosphate (PIP2) | Avanti Polar Lipids | Cat. #: 840046 | For nanodisc reconstitution (see Materials and methods section) |
| Antibody | Rho anti-1D4 (mouse monoclonal) | Santa Cruz Biotechnologies | Cat. #: sc-57432; RRID:AB_785511 | (0.5 µg/mL) |
| Antibody | THE anti-His (mouse monoclonal) | GenScript | Cat. #: A00186; RRID:AB_914704 | (0.33 µg/mL) |
| Antibody | M1 anti-FLAG (mouse monoclonal) | Sigma | Cat. #: F3040; RRID:AB_439712 | (0.5 µg/mL) |
| Antibody | IRDye 800 goat anti-mouse secondary | LI-COR | Cat. #: 926–32210; RRID:AB_621842 | (1:15,000) |

*Continued on next page*

*Continued*

| Reagent type (species) or resource | Designation | Source or reference | Identifiers | Additional information |
|---|---|---|---|---|
| Peptide, recombinant protein | Dylight Neutravidin-800 | Invitrogen | Cat. #: 22853 | (1:10,000) |
| Other | CNBr-activated sepharose | Cytiva | Cat. #: 17043001 | To generate GFP nanobody resin (see Materials and methods section) |
| Other | Streptavidin paramagnetic particles | Promega | Cat. #: Z5481 | For bead binding competition experiments (see Materials and methods section) |
| Strain, strain background (*Escherichia coli*) | BL21(DE3)-RIL | Agilent | Cat. #: 230245 | |
| Cell line (*Spodoptera frugiperda*) | Sf9 cells | Expression Systems | Cat. #: 94-001F; RRID:CVCL_0549 | |
| Cell line (human) | Expi293 cells | Thermo | Cat. #: A14527; RRID:CVCL_D615 | |
| Cell line (human) | Freestyle293 cells | Thermo | Cat. #: R79007; RRID:CVCL_D603 | |
| Cell line (human) | HEK293T cells (Fzd1/2/4/5/7/8 KO) | PMID:28733458 | | Boutros lab |
| Sequence-based reagent | Tspan12 siRNA duplex #1 | Sigma | Cat. #: VC30002 | 5'-GCUUAUCUUUGCCUUCUCCTT-3' and 5'-GGAGAAGGCAAAGAUAAGCTT-3' |
| Sequence-based reagent | Tspan12 siRNA duplex #2 | Sigma | Cat. #: VC30002 | 5'-AUGAGGGACUACCUAAAUATT-3' and 5'-UAUUUAGGUAGUCCCUCAUTT-3' |
| Sequence-based reagent | Control siRNA | Sigma | Cat. #: SIC001 | |
| Recombinant DNA reagent | Super8xTOPFLASH | Addgene | RRID:Addgene_12456 | Moon lab |
| Commercial assay or kit | Dual-Light reporter assay system | Applied Biosystems | Cat. #: T1005 | |
| Software, algorithm | Prism 9 | GraphPad Software, LLC | RRID:SCR_002798 | |
| Software, algorithm | AlphaFold2 | PMID:34265844 | RRID:SCR_025454 | |
| Software, algorithm | AlphaFold Multimer v2 | DOI: 10.1101/2021.10.04.463034 | | |
| Software, algorithm | ColabFold | PMID:35637307 | RRID:SCR_025453 | |
| Software, algorithm | ChimeraX 1.6 | PMID:28710774 | RRID:SCR_015872 | |
| Software, algorithm | cryoSPARC v4 | Structura Biotechnology | RRID:SCR_016501 | |

## Cell lines

HEK293T cells with Fzd1/2/4/5/7/8 knocked out were provided by Michael Boutros, who authenticated the parental HEK293T cell line by single nucleotide polymorphism-based authentication (*Voloshanenko et al., 2017*). Freestyle293 cells (Thermo), Expi293 cells (Thermo), and Sf9 cells (Expression Systems) were purchased directly from vendors who provided a certificate of analysis, so we have not sent these cells for authentication. All four cell lines tested negative for mycoplasma contamination via PCR (abm).

## Protein constructs, expression, and purification

Human Tspan12 (DNASU) truncated after the fourth transmembrane domain (ΔC, residues 1–252) and Tspan12 with the LEL replaced by that of TSPAN11 (Tspan12-LEL11, Addgene plasmid #115785 from Harald Junge; *Lai et al., 2017*) were C-terminally tagged with the Rho-1D4 antibody recognition sequence TETSQVAPA. Full-length Tspan12 (1-305) and derived point mutants, also with a C-terminal TETSQVAPA, were cloned into pTT5 (NRC) for purification from Expi293 cells. Mouse Fzd4 (42–537) and human LRP6 (20–1439) were preceded by an N-terminal hemagglutinin signal peptide (HA$_{sp}$) followed by FLAG tag. For the split GFP tethered dimers (*Bruguera et al., 2022*), the sequence of

HA$_{sp}$-FLAG-mFzd4ΔC(42-513) was followed by a 1x GS linker, the HRV 3C recognition sequence, a 1x GGTS linker, the split GFP 1–10 OPT sequence engineered by the Waldo group (*Cabantous et al., 2005*), a gift from Steven Boxer (*Deng and Boxer, 2018*), a 2x GS linker, and the C-terminal PDZ ligand from Fzd4 (sequence ETVV). Tspan12ΔC (1–252) was followed by a 6xHis tag, a 1x GS linker, the HRV 3C recognition sequence, a 2x GS linker, and the GFP 11 M3 sequence (*Cabantous et al., 2005*). Tspan12ΔC-1D4, Tspan12-LEL11-1D4, HA-FLAG-Fzd4, HA-FLAG-Fzd4ΔC-3C-GFP1-10OPT-ETVV, and Tspan12ΔC-His-3C-GFP11 were inserted into pVL1393 (Expression Systems) or pFastBac1 (Invitrogen) by Gibson Assembly (New England Biolabs) for expression in insect cells. LRP6 and its chaperone MESD were inserted into the pEZT-BM vector (Addgene #74099) for expression in Freestyle293 cells. The Tspan12 LEL (residues 118–218), N-terminally tagged with maltose binding protein and the 3C protease recognition sequence, was inserted into the pAcGP67a transfer vector (BD Biosciences). Virus was produced and amplified according to the manufacturer's instructions.

For protein expression, all lysis, wash, and affinity column buffers included the protease inhibitors 0.15 µM aprotinin, 1 µM E-64, and 1 µM leupeptin, and 200 µM phenylmethylsulfonyl fluoride. Buffers for constructs including the transmembrane domain additionally included 60 µM *N-p*-Tosyl-L-phenylalanine chloromethyl ketone and 60 µM $N_{\alpha}$-Tosyl-L-lysine chloromethyl ketone. All steps were performed at 4°C.

For receptor preparations of Tspan12, Fzd4, and Tspan12/Fzd4-sGFP, Sf9 cells (Expression Systems) were infected at a density of $3×10^6$ cells/mL with 1:300 vol/vol virus and harvested after 48 hr. WT and mutant Tspan12 were produced for binding studies in Expi293 cells transfected with PEIpro (Polyplus) according to the manufacturer's instructions, enhanced 16 hr later with 10 mM sodium butyrate (Sigma), and harvested 48 hr post-transfection. Sf9 or Expi293 cells were lysed using nitrogen cavitation (Parr Instrument Company) at 650 psi for 30 min in 20 mM HEPES pH 8.0, 65 mM NaCl (for Sf9) or 10 mM NaCl (for Expi293), 1 mM EDTA, and 10 mM iodoacetamide. Nuclei and cell debris were pelleted at 1000 × *g* for 15 min, and the resulting supernatant was centrifuged at 200,000 × *g* in an ultracentrifuge for 40 min. Pelleted membranes were Dounce homogenized 30× to resuspend into high-salt buffer (50 mM HEPES pH 8, 300 mM NaCl) and centrifuged at 200,000 × *g* for 40 min. Membranes were Dounce homogenized 30× into low-salt buffer (50 mM HEPES pH 8, 100 mM NaCl) and frozen dropwise in liquid nitrogen, then stored at –80°C until use.

Tspan12 membranes were thawed and adjusted to 5mg/mL protein (determined by Bradford assay), 20mM HEPES 8, 100mM NaCl, 10% (vol/vol) glycerol, 2.5mM EGTA, 2.5mM EDTA, 1% (wt/vol) *n*-dodecyl-β-D-maltopyranoside (DDM; Anatrace), 0.1% (wt/vol) GDN (Anatrace), and 0.1% (wt/vol) cholesteryl hemisuccinate (CHS; Anatrace). The membranes were stirred for 2hr at 4°C, then centrifuged at 200,000× *g* for 1hr. The supernatant was filtered through a 0.2µm filter prior to loading onto Rho-1D4 antibody resin (2mL resin per L culture), and beads were washed by gravity with 5 column volumes (CVs) each of 20mM HEPES pH 8, 1mM EDTA, 5% glycerol, plus (a) 300mM NaCl, 0.07% DDM, 0.007% CHS, 0.03% GDN; (b) 100mM NaCl, 0.04% DDM, 0.004% CHS, 0.06% GDN; (c) 100mM NaCl, 0.01% DDM, 0.001% CHS, 0.09% GDN; (d) 100mm NaCl, 0.1% GDN. The column was washed with one CV of elution buffer (20mM HEPES pH 8, 100mM NaCl, 1mM EDTA, 10% glycerol, 0.02% GDN, 200µM TETSQVAPA peptide [GenScript]), capped, and batch eluted rotating overnight with 1 additional CV elution buffer at 4°C. Eluate was collected 16hr later, concentrated in a 30kDa cutoff concentrator (Amicon) and further purified by size exclusion on a Superose 6 increase 10/300 column (Cytiva) in 20mM HEPES pH 8, 100mM NaCl, 1mM EDTA, 5% glycerol, 0.01% GDN. Analysis by SDS-PAGE informed pooling of pure, monomeric fractions, which were concentrated to 2–5mg/mL and frozen at –80°C.

Fzd4 and Tspan12/Fzd4 split-GFP were purified as above and as previously described (*Bruguera et al., 2022*; *Mahoney et al., 2022*). Briefly, the purification is similar to that detailed above with the following modifications: M1 anti-FLAG resin was used instead of 1D4 resin; binding and wash buffers did not contain EDTA or EGTA and were supplemented with 3 mM CaCl$_2$, and elution buffer instead contained DYKDDDDK peptide (GenScript) and 5 mM EGTA. Additionally, for LRP6 and Fzd4 alone, wash and elution buffers did not contain glycerol or EDTA, and receptor was not exchanged into GDN (all wash buffers contained 0.1% DDM and 0.001% CHS; elution and SEC buffers contained 0.03% DDM and 0.003% CHS).

FLAG-LRP6(20–1439) was co-expressed in Freestyle293 cells with the chaperone MESD and purified as previously described (*Bruguera et al., 2022*). Briefly, the purification is similar to FLAG-tagged

Fzd4 above with the following modifications: after binding, the anti-FLAG resin was washed with 10 CVs of low pH buffer (50 mM sodium acetate [pH 5.0], 150 mM NaCl, 0.1% DDM, 0.01% CHS, and 2 mM CaCl$_2$) to remove bound MESD, and 10 CVs of ATP wash (low salt buffer supplemented with 5 mM ATP, 20 mM MgCl$_2$, and 50 mM KCl) to remove bound HSP 70, before elution with DYKDDDDK peptide and EGTA as above. The final SEC buffer included 2 mM CaCl$_2$ instead of EDTA.

MBP-3C-Norrin(33–133)-1D4 (*Bruguera et al., 2022*; *Mahoney et al., 2022*), Fzd4 extracellular domain (CRD and linker, residues 42–203, C-terminally His$_6$-tagged) (*Bang et al., 2018*) and LRP6 E1E2 (residues 20–630, C-terminally His$_{10}$-tagged) (*Ahn et al., 2011*) in the (BD Biosciences) transfer vector were purified from baculovirus-infected Sf9 cells according to the previously published works. For binding experiments, purified MBP-3C-Norrin was biotinylated with EZ-link NHS-PEG4-Biotin (Thermo) at 1:1 molar ratio and re-purified by SEC. Monomeric Norrin (MBP-3C-Norrin C93A/C95A/C131A) was purified identically to dimeric Norrin (*Figure 4—figure supplement 4*). MBP-tagged Tspan12 LEL, also in pAcGP67, was purified similarly in the same buffer (20 mM HEPES pH 8, 300 mM NaCl, 1 mM EDTA, 5% glycerol). For untagged LEL, the LEL was eluted from the amylose resin with 3C protease and minimally concentrated before injection onto a HiLoad 26/600 Superdex 200 pg column (Cytiva). Pooled fractions were concentrated in a 3 kDa cutoff spin concentrator (Amicon) to 0.3 mg/mL for storage at –80°C.

The purification of MBP-tagged DEP domain of mouse Dishevelled2 (residues 416–510) from BL21(DE3)-RIL cells was performed as previously described (*Mahoney et al., 2022*).

The His$_6$-tagged GFP nanobody sequence (*Kubala et al., 2010*) was cloned into pET26(+), expressed in the periplasm of *E. coli* BL21(DE3)-RIL, and purified according to previously published protocols (*Pardon et al., 2014*). The nanobody was coupled to CNBr-activated Sepharose (Cytiva) according to the manufacturer's instructions.

His$_6$-tagged MSP1D1 and MSP1E3D1 (Addgene plasmids #20061 and #20066) were expressed in *E. coli* BL21(DE3)-RIL and purified (*Bayburt et al., 2002*; *Ritchie et al., 2009*) and biotinylated using EZ-link NHS-PEG4-Biotin (Thermo) as previously described (*Bruguera et al., 2022*; *Mahoney et al., 2022*).

## Nanodisc reconstitution

Receptors were inserted into nanodiscs as previously described (*Bruguera et al., 2022*; *Mahoney et al., 2022*; *Ritchie et al., 2009*; *Whorton et al., 2007*). Briefly, lipids (16:0-18:1 phosphatidyl choline [POPC], 16:0–18:1 phosphatidylglycerol (POPG) and 18:0-20:4 brain phosphatidylinositol-4,5-bisphosphate (PIP$_2$), Avanti Polar Lipids) and cholesterol (Sigma) were purchased pre-dissolved in organic solvent, and opened stocks were stored under argon at –20°C for <3 months (POPC, POPG, PIP$_2$) or <1 month (cholesterol). Lipids were mixed at a 48:32:20 POPC:POPG:cholesterol (or 75:5:20 POPC:PIP$_2$:cholesterol, for DEP binding [*Mahoney et al., 2022*]) molar ratio and dried under a stream of argon and then under vacuum for 1 hr. Dried lipids were resuspended in HNE (20 mM HEPES pH 8.0, 100 mM NaCl, 1 mM EDTA) supplemented with 50 mM sodium cholate. HNE buffer, receptor, and MSP were added to reach final concentrations of 18 mM sodium cholate, 6 mM lipid, 0.1 mM MSP1D1 or 0.07 mM MSP1E3D1, and 5 µM receptor monomer or receptor dimer. After incubation on ice for 1 hr, detergent was removed with Bio-Beads (Bio-Rad, 83 mg beads per nmol of lipids) overnight at 4°C.

Nanodiscs were further purified by size exclusion (Superose 6 Increase 10/300, Cytiva) in HNE buffer followed by M1 anti-FLAG (Fzd4), Rho anti-1D4 (Tspan12), or GFP nanobody (Tspan12/Fzd heterodimer; eluted with 3C protease) affinity chromatography. Wash and elution buffer for preparative samples consisted of HNE+0.1% bovine serum albumen (BSA; Sigma). BSA was omitted to enable clear analysis by SDS-PAGE (i.e., in *Figure 1—figure supplement 1E* and *Figure 4—figure supplement 1E*).

The eluted nanodiscs were run on SDS-PAGE along with a standard curve of known amounts of MSP, and concentration was thus quantified using densitometry in ImageJ. Nanodiscs were stored on ice for up to 2 weeks.

For stoichiometry measurements, the concentration of MSP and receptor within each relevant sample was determined by quantitative western blot as described (*Bruguera et al., 2022*). Briefly, serial dilutions of reference proteins alongside nanodisc samples, diluted to be within the linear range of detection for the blotted protein, were loaded on SDS-PAGE gels and transferred to nitrocellulose. For Tspan12 receptor alone in nanodiscs, 0.04–0.32 picomoles of Tspan12-1D4 and 0.25–2 picomoles

of 7xHis-MSP1D1 were probed by Rho anti-1D4 and THE anti-His (GenScript) antibodies respectively. For the Tspan12/Fzd4 heterodimer, 0.125–1 picomoles of Tspan12-6xHis, 0.01–0.08 picomoles of FLAG-Fzd4 were probed by THE anti-His (GenScript) and M1 anti-FLAG antibodies, respectively; and 0.25–2 picomoles of MSP1D1-biotin was detected by DyLight 800-conjugated Neutravidin (Invitrogen), as His-tagged 3C protease runs similarly to MSP1D1 on a gel. All other samples were detected with IRDye 800 Goat-anti Mouse IgG (LI-COR). For each of the two nanodisc species (Tspan12 alone and Tspan12/Fzd4 heterodimer), all proteins within three independently reconstituted samples were measured three times each in separate western blots.

## Biolayer interferometry

Kinetic and steady-state binding affinities were measured by BLI as previously described (*Bruguera et al., 2022*; *Mahoney et al., 2022*) using Octet RED384 (Sartorius) or GatorPrime (Gator Bio) instruments at 25°C at 1000 rpm shaking, in 20 mM HEPES pH 7.4, 150 mM NaCl, 1 mM EDTA, 0.1% BSA. Binding buffer for experiments that included LRP6 (*Figure 5—figure supplement 1B and C*) also included 3 mM CaCl$_2$. Binding experiments between soluble proteins only (i.e., the Tspan12 LEL and Norrin, ±CRDL) additionally included 0.05% Tween 20 to minimize non-specific binding. Streptavidin-coated biosensors were loaded with 20 nM biotinylated nanodiscs for 5 min (yielding an interference shift between 1.5 and 2.5 nm) prior to binding.

For Norrin-binding experiments, 100 nM MBP-Norrin in binding buffer was incubated with 3C protease for 30 min at room temperature to cleave the MBP tag before binding. SOS (potassium salt) was obtained from Santa Cruz Biotechnology. For DEP binding in the presence of Norrin, biosensor-immobilized Fzd4±Tspan12 receptors were pre-bound to 10 nM Norrin for 30 min before DEP binding was conducted in buffer containing 10 nM Norrin.

After preliminary processing (Savitzky-Golay filtering, signal subtraction of control conditions, i.e., ligand binding to receptor-less nanodiscs, which was run in parallel) in the Octet Data Analysis 10.0 or Gator 1.7 software, curve fitting was performed in Prism (GraphPad). Steady-state binding affinity values and Hill slopes were obtained by fitting equilibrium data (signal plateau value) to a one-site model of specific binding with variable Hill slope. When multiple concentrations of analyte were measured, the association rate constant $K_{on}$ was determined as the slope of the line fit through a plot of the observed association rate constant ($K_{obs}$, obtained from one-phase exponential association fits to each association trace) vs. ligand concentration. Dissociation traces were fit to one-phase exponential decay curves to obtain the dissociation rate constant $K_{off}$, and the mean $K_{off}$ was calculated using dissociation data for which the $R^2$ was >0.8 (i.e., discarding the low concentration conditions with low signal). For experiments in which only one ligand concentration was measured (i.e., mutant Tspan12 and Norrin-binding measurements in *Figure 2*, *Figure 2—figure supplement 3*, and *Figure 2—figure supplement 4*), the $K_{on}$ and $K_{off}$ were determined using the 'Association then dissociation' model in Prism.

Each binding experiment was performed in triplicate, using at least two independent preparations each of receptor and ligand.

## AlphaFold predictions

The sequence for Tspan12 (full-length) alone or with one or two copies of Norrin (residues 25–133) were input into AlphaFold 2 (*Jumper et al., 2021*) or AlphaFold Multimer v2 (*Evans et al., 2022*) using ColabFold (*Mirdita et al., 2022*) via ChimeraX (1.6, daily build from October 23, 2022) (*Goddard et al., 2018*). The following PDB structures were used as templates for Tspan12: 6wvg_A, 6wvg_B, 6k4j_A, 5tcx_A, 7rdb_A, 7rdb_H, 2m7z_A, 6wvg_A, 6wvg_B; and for Norrin: 5bpu_A, 5bqe_B, 5bq8_A, 5aej_A, 4jph_B, 5hk5_F, 2kd3_A, 4nt5_A, 4x1j_B, 6l6r_C, 6l6r_D, 2k8p_A, 4x1j_A, 6p57_A, 7fih_Y, 5cl1_A, 4yu8_A, 4ay9_B, 4mqw_B, 4mqw_E. The top-ranked structure (sorted by pLDDT for Tspan12 alone, or pTMscore for Tspan12+Norrin complexes) was relaxed with Amber (*Eastman et al., 2017*) and used for further studies.

## Bead binding competition experiments

Streptavidin paramagnetic particles (Promega), pre-equilibrated in wash buffer (20 mM HEPES pH 7.4, 150 mM NaCl, 1 mM EDTA, 0.2% BSA), were pre-bound to Tspan12ΔC (reconstituted in biotinylated MSP1D1) or receptorless nanodiscs. After 30 min at 25°C, the beads were quenched with biotin and

washed three times with wash buffer. MBP-Norrin at 32 nM was premixed with 0, 0.1, 0.4, 1.6, 25, or 100 μM LRP6 E1E2 or Fzd4 CRDL and added to the beads, which were at a final concentration of 0.2 mg/mL paramagnetic particles and 50 nM Tspan12ΔC; these conditions were chosen such that, in the absence of competitor, approximately half of the Norrin would be bound. Tubes were rotated for 16 hr at 4°C, then beads were washed three times with ice-cold wash buffer and protein was eluted with 1× non-reducing Laemmli sample buffer (non-reducing) at room temperature for 15 min. Three replicates per condition were loaded on an SDS-PAGE gel, transferred to nitrocellulose, and analyzed by western blot with anti-Rho 1D4 antibody, followed by goat anti-mouse 800 (LI-COR). Background-subtracted band intensities, which were verified to be in the linear range of detection using a standard curve loaded on the same gel, were quantified using Odyssey software (LI-COR). Data were analyzed in Prism (GraphPad) using the following system of equations (**Wang, 1995**), which model the fraction of Norrin bound vs. competitor concentration (Fzd4 CRDL or LRP6 E1E2):

$$h = KA + KB + A + X - R$$
$$k = [KB*(A-R)] + [KA*(X-R)] + (KA*KB)$$
$$l = -1*(KA*KB*R)$$
$$t = \arccos[(-2*(h\char`\^3) + 9*h*k - 27*l)/(2*\mathrm{sqrt}((h\char`\^2 - 3k)\char`\^3))]$$
$$Y = [(2*(\mathrm{sqrt}(h\char`\^2 - 3k))*(\cos(t/3)) - h)/(3*KA + (2*(\mathrm{sqrt}(h\char`\^2 - 3k))*(\cos(t/3)) - h))]$$

where

X = competitor concentration
Y = Norrin bound (as a fraction of total)
A = total Norrin binding sites = 64 nM
R = total Tspan12 concentration = 50 nM
KA = equilibrium Norrin-Tspan12 dissociation constant = 10.4 nM
KB = equilibrium Norrin-competitor dissociation constant (to be fit)
For experiments where only bead-bound Norrin was measured (instead of Bead+Sup, to obtain fraction bound), Y was normalized to the calculated fraction of Norrin bound in the absence of competitor, defined as (A+KA+R-sqrt((-A-KA-R)^2–4*A*R))/(2*A).

For modeling SOS-mediated inhibition of Norrin-Tspan12 binding assayed by BLI, the $IC_{50}$ was converted to $K_i$ using the Cheng-Prusoff equation.

## Negative stain electron microscopy

To analyze the oligomeric state of Norrin, MBP-Norrin (WT or C93A/C95A/C131A) was diluted to 100 nM in purification buffer (20 mM HEPES pH 8, 300 mM NaCl, 1 mM EDTA, 5% glycerol) and applied to glow-discharged CF-300Cu grids (Electron Microscopy Sciences), which were washed once with buffer (20 mM HEPES pH 8, 100 mM NaCl, 1 mM EDTA) and twice with 1% uranyl acetate stain before drying and imaging on a 100 kV Morgagni electron microscope equipped with an Orius CCD camera (Gatan) at 50,000×. Particle picking and 2D class averaging was performed in cryoSPARC v4 (Structural Biotechnology Inc).

## Cell-binding experiments

Expi293 cells (Thermo Fisher) were maintained in Expi293 media (Thermo Fisher) and transfected at a density of $3\times10^6$ cells/mL using ExpiFectamine293 (Thermo Fisher) according to the manufacturer's instructions. Cells were transfected with untagged Tspan12 in pTT5 (NRC), FLAG-tagged Fzd4 in pcDNA3.1 (Thermo Fisher), or both Tspan12 and Fzd4 (300 ng each receptor per mL of cells), with additional empty vector to bring total DNA transfected to 1 μg/mL of cells. 48 hr post-transfection, cells were transferred to a 96-well v-bottom plate and incubated with 0, 0.032, 0.1, 0.32, 1, or 3.2 nM purified MBP-Norrin in binding buffer (20 mM HEPES pH 7.4, 150 mM NaCl, 1 mM EDTA, 0.2% BSA) for 10 min at 23°C, shaking. Cells were collected by centrifugation and washed twice with binding buffer and incubated with Alexa Fluor 647-conjugated anti-rho 1D4 antibody in binding buffer for 15 min at room temperature. Cells were washed once with binding buffer and resuspended in PBS prior to detection with an Accuri C6 Plus flow cytometer (BD). Data were exported to Prism (GraphPad) for quantification of median fluorescence intensity and statistical analysis.

## β-Catenin transcriptional reporter assay and Fzd knockout cell line

HEK293T cells with Fzd1/2/4/5/7/8 knocked out (*Voloshanenko et al., 2017*) were maintained in DMEM (Gibco) supplemented with 10% fetal bovine serum (Gemini). In a six-well plate, cells were seeded at 200,000 cells/well and immediately transfected with 30 picomoles of control siRNA (Sigma SIC001), or 15 picomoles each of two previously validated (*Otomo et al., 2014*) Tspan12 siRNA duplexes (#1: 5'-GCUUAUCUUUGCCUUCUCCUU-3' and 5'-GGAGAAGGCAAAGAUAAGCUU-3'; #2: 5'-AUGAGGGACUACCUAAAUAUU-3' and 5'-UAUUUAGGUAGUCCCUCAUU-3') (Sigma) using Lipofectamine RNAiMax (Thermo Fisher) diluted in OptiMEM (Thermo Fisher) according to the manufacturer's instructions, with media replaced after 16 hr. Two days post-transfection, cells were seeded at 12,000 cells/well in white 96-well plates (PerkinElmer) and transfected 4 hr later with receptor vectors in pcDNA (0.1 ng/well FLAG-Fzd4_pcDNA, untagged Tspan12_pTT5 amounts as indicated), Super8xTOPFLASH (Addgene plasmid #12456, 80 ng/well) and LacZ under a CMV promoter (20 ng/well) using Lipofectamine 2000 (Invitrogen). Media was replaced 16–20 hr later with DMEM alone or supplemented with purified Norrin, which was cleaved from MBP with 3C protease prior to addition to cells. Cells were lysed 22–26 hr later and assayed for luciferase and β-galactosidase activity using the Dual-Light system (Invitrogen) according to the manufacturer's instructions on a BioTek Synergy2 plate reader. Luciferase signal was normalized to β-galactosidase signal; all TOPFLASH values are reported as fold change over the basal signal (defined as 0 ng Tspan12, 0 nM Norrin, in cells transfected with control siRNA).

## Acknowledgements

ESB was supported by an NIH predoctoral fellowship (F31 EY031947) and a Stanford Graduate Fellowship, and JPM was supported by an NIH postdoctoral fellowship (F32 GM126642). This work was supported by an NIH grant (R35 GM131747 to WW). We thank Roel Nusse, Sabine Pokutta, Kaavya Krishna Kumar, and Brian Kobilka of Stanford University, as well as Jon-Michael Knapp of Luminint Consulting, for comments on the manuscript. We thank Samantha Gumbin for advice on siRNA experiments. We are grateful to Liz Montabana of the Stanford Cryo-EM Center for assistance with electron microscopy, and to Sharon Pitteri and Abel Bermudez of the Canary Center Proteomics Resource Facility for access to BLI instrumentation.

## Additional information

### Funding

| Funder | Grant reference number | Author |
| --- | --- | --- |
| National Institutes of Health | GM131747 | William I Weis |
| National Institutes of Health | EY031947 | Elise S Bruguera |
| National Institutes of Health | GM126642 | Jacob P Mahoney |

The funders had no role in study design, data collection and interpretation, or the decision to submit the work for publication.

### Author contributions

Elise S Bruguera, Conceptualization, Formal analysis, Funding acquisition, Investigation, Visualization, Methodology, Writing – original draft, Writing – review and editing; Jacob P Mahoney, Conceptualization, Resources, Supervision, Methodology, Writing – review and editing; William I Weis, Conceptualization, Resources, Supervision, Funding acquisition, Methodology

### Author ORCIDs

Elise S Bruguera ⓘ https://orcid.org/0000-0003-1983-3013
William I Weis ⓘ https://orcid.org/0000-0002-5583-6150

Joint Public Review: https://doi.org/10.7554/eLife.96743.3.sa1
Author response https://doi.org/10.7554/eLife.96743.3.sa2

---

## Additional files

### Supplementary files
Supplementary file 1. Table of kinetic constants and steady-state affinities quantified by biolayer interferometry.

MDAR checklist

### Data availability
Source data files containing the numerical data used to generate figures 1-5 have been provided.

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
