## [Editor Report · eLife Assessment]

This is a **fundamental** study that addresses the key question of how the tetraspanin Tspan12 functions biochemically as a co-receptor for Norrin to initiate β-catenin signaling. The strength of the work lies in the rigorous and **compelling** binding analyses involving various purified receptors, co-receptors, and ligands, as well as molecular modeling by AlphaFold that was subsequently validated by an extensive series of mutagenesis experiments. The study advances the field by providing a novel mechanism of co-receptor function and shedding new light on how signaling specificity is achieved in the complex Wnt/Norrin signaling system.

---

## [Referee Report · Joint Public Review]

Though the Norrin protein is structurally unrelated to the Wnt ligands, it can activate the Wnt/β-catenin pathway by binding to the canonical Wnt receptors Fzd4 and Lrp5/6, as well as the tetraspanin Tspan12 co-receptor. Understanding the biochemical mechanisms by which Norrin engages Tspan12 to initiate signaling is important, as this pathway plays an important role in regulating retinal angiogenesis and maintaining the blood-retina-barrier. Numerous mutations in this signaling pathway have also been found in human patients with ocular diseases. The overarching goal of the study is to define the biochemical mechanisms by which Tspan12 mediates Norrin signaling. Using purified Tspan12 reconstituted in lipid nanodiscs, the authors conducted detailed binding experiments to document the direct, high-affinity interactions between purified Tspan12 and Norrin. To further model this binding event, they used AlphaFold to dock Norrin and Tspan12 and identified four putative binding sites. They went on to validate these sites through mutagenesis experiments. Using the information obtained from the AlphaFold modeling and through additional binding competition experiments, it was further demonstrated that Tspan12 and Fzd4 can bind Norrin simultaneously, but Tspan12 binding to Norrin is competitive with other known co-receptors, such as HSPGs and Lrp5/6. Collectively, the authors proposed that the main function of Tspan12 is to capture low concentrations of Norrin at the early stage of signaling, and then "hand over" Norrin to Fzd4 and Lrp5/6 for further signal propagation. Overall, the study is comprehensive and compelling, and the conclusions are well supported by the experimental and modeling data.

---

## [Author Response]

The following is the authors’ response to the original reviews.

**Reviewer #1 (Public Review):**
Though the Norrin protein is structurally unrelated to the Wnt ligands, it can activate the Wnt/βcatenin pathway by binding to the canonical Wnt receptors Fzd4 and Lrp5/6, as well as the tetraspanin Tspan12 co-receptor. Understanding the biochemical mechanisms by which Norrin engages Tspan12 to initiate signaling is important, as this pathway plays an important role in regulating retinal angiogenesis and maintaining the blood-retina-barrier. Numerous mutations in this signaling pathway have also been found in human patients with ocular diseases. The overarching goal of the study is to define the biochemical mechanisms by which Tspan12 mediates Norrin signaling. Using purified Tspan12 reconstituted in lipid nanodiscs, the authors conducted detailed binding experiments to document the direct, high-affinity interactions between purified Tspan12 and Norrin. To further model this binding event, they used AlphaFold to dock Norrin and Tspan12 and identified four putative binding sites. They went on to validate these sites through mutagenesis experiments. Using the information obtained from the AlphaFold modeling and through additional binding competition experiments, it was further demonstrated that Tspan12 and Fzd4 can bind Norrin simultaneously, but Tspan12 binding to Norrin is competitive with other known co-receptors, such as HSPGs and Lrp5/6. Collectively, the authors proposed that the main function of Tspan12 is to capture low concentrations of Norrin at the early stage of signaling, and then "hand over" Norrin to Fzd4 and Lrp5/6 for further signal propagation. Overall, the study is comprehensive and compelling, and the conclusions are well supported by the experimental and modeling data.Strengths:• Biochemical reconstitution of Tspan12 and Fzd4 in lipid nanodiscs is an elegant approach for testing the direct binding interaction between Norrin and its co-receptors. The proteins used for the study seem to be of high purity and quality.• The various binding experiments presented throughout the study were carried out rigorously. In particular, BLI allows accurate measurement of equilibrium binding constants as well as on and off rates.• It is nice to see that the authors followed up on their AlphaFold modeling with an extensive series of mutagenesis studies to experimentally validate the potential binding sites. This adds credence to the AlphaFold models.• Table S1 is a further testament to the rigor of the study.• Overall, the study is comprehensive and compelling, and the conclusions are well supported by the experimental and modeling data.Suggestions for improvement:• It would be helpful to show Coomassie-stained gels of the key mutant Norrin and Tspan12 proteins presented in Figures 2E and 2F.

We have included Stain-Free SDS-PAGE gels from the purification of the Norrin and Tspan12 mutants in a new Figure S4.

• Many Norrin and Tspan12 mutations have been identified in human patients with FEVR. It would be interesting to comment on whether any of the mutations might affect the NorrinTspan12 binding sites described in this study.

Thank you for this suggestion. We have inspected human mutation databases gnomAD, ClinVar, and HGMD for known mutations in the predicted Tspan12-Norrin binding interface and their occurrence in human patients with FEVR or Norrie disease.

While a number of Tspan12 residues that we predict to interact with Norrin are impacted by rare mutations in humans (e.g., L169M, E170V, E173K, D175N, E196G, S199C, as found in the gnomAD database), these alleles are of unknown clinical significance (as found in ClinVar or HGMD databases). It is possible that mutations that slightly weaken the Norrin-Tspan12 interface may not produce a strong phenotype, especially given the avidity we expect from this system. By our examination, the missense variants of clinical significance that have been found in the Tspan12 LEL would be expected to destabilize the protein (i.e., mutations to or from cysteine or proline, or mutations to residues involved in packing interactions within the LEL fold), and therefore these mutations may produce a disease phenotype by impacting Tspan12 protein expression levels.

Several Norrin mutations that are associated with Norrie disease, FEVR, or other diseases of the retinal vasculature have been found in the predicted Tspan12 binding site. For example, Norrin mutations at positions L103 (L103Q, L103V), K104 (K104N, K104Q), and A105 (A105T, A105P, A105E, A105S, A105T, A105V) have been found in patients, all of which may disrupt binding to Tspan12. However, the deleterious effect of K104 mutations on Norrin-stimulated signaling could also be explained by a weakened Norrin-Fzd4 binding interface. Norrin mutations at R115 (R115L and R115Q), as well as R121 (R121L, R121G, R121Q, and R121W) have also been found in patients with various diseases of the retinal vasculature. Additionally, the Norrin mutation T119P has been found in patients with Norrie disease, but we would expect this mutation to destabilize Norrin in addition to disrupting the Tspan12 binding site.

While we commented briefly on mutations R115L and R121W in the original draft (page 5, paragraphs 4 and 1, respectively), we have updated the manuscript with more comments on disease-associated mutations to the predicted Tspan12 binding site on Norrin (page 5, first partial paragraph; page 9, first partial paragraph).

• Some of the negative conclusions (e.g. the lack of involvement of Tspan12 in the formation of the Norrin-Lrp5/6-Fzd4-Dvl signaling complex) can be difficult to interpret. There are many possible reasons as to why certain biological effects are not recapitulated in a reconstitution experiment. For instance, the recombinant proteins used in the experiment may not be presented in the correct configurations, and certain biochemical modifications, such as phosphorylation, may also be missing.

We agree that different Tspan12 and Fzd4 stoichiometries, lipid compositions, and posttranslational modifications could impact the results of our study, and that it is important to mention these possibilities. We have added these caveats to the discussion section (page 10, last paragraph).

**Reviewer #2 (Public Review):**
This is an interesting study of high quality with important and novel findings. Bruguera et al. report a biochemical and structural analysis of the Tspan12 co-receptor for norrin. Major findings are that Norrin directly binds Tspan12 with high affinity (this is consistent with a report on BioRxiv: Antibody Display of cell surface receptor Tetraspanin12 and SARS-CoV-2 spike protein) and a predicted structure of Tspan12 alone or in complex with Norrin. TheNorrin/Tspan12 binding interface is largely verified by mutational analysis. An interaction of the Tspan12 large extracellular loop (LEL) with Fzd4 cannot be detected and interactions of fulllength Tspan12 and Fzd4 cannot be tested using nano-disc based BLI, however, Fzd4/Tspan12 heterodimers can be purified and inserted into nanodiscs when aided by split GFP tags. An analysis of a potential composite binding site of a Fzd4/Tspan12 complex is somewhat inconclusive, as no major increase in affinity is detected for the complex compared to the individual components. A caveat to this data is that affinity measurements were performed for complexes with approximately 1 molecule Tspan12 and FZD4 per nanodisc, while the composite binding site could potentially be formed only in higher order complexes, e.g., 2:2 Fzd4/Tspan12 complexes. Interestingly, the authors find that the Norrin/Tspan12 binding site and the Norrin/Lrp6 binding site partially overlap and that the Lrp6 ectodomain competes with Tspan12 for Norrin binding. This result leads the authors to propose a model according to which Tspan12 captures Norrin and then has to "hand it off" to allow for Fzd4/Lrp6 formation. By increasing the local concentration of Norrin, Tspan12 would enhance the formation of the Fzd4/Lrp5 or Fzd4/Lrp6 complex.

Thank you for pointing out the BioRxiv report showing Norrin-Tspan12 LEL binding. We have cited this in the introduction of our revised manuscript (page 2, paragraph 3).

The experiments based on membrane proteins inserted into nano-discs and the structure prediction using AlphaFold yield important new insights into a protein complex that has critical roles in normal CNS vascular biology, retinal vascular disease, and is a target for therapeutic intervention. However, it remains unclear how Norrin would be "handed off" from Tspan12 or Tspan12/Fzd4 complexes to Fzd4/Lrp6 complexes, as the relatively high affinity of Norrin to Fzd4/Tspan12 dimers likely does not favor the "handing off" to Fzd4/Lrp6 complexes.

While the Fzd4-Tspan12 interaction is strong, our data suggest that Fzd4 and Tspan12 bind Norrin with negative cooperativity, suggesting that Fzd4 binding may enhance Norrin-Tspan12 dissociation to facilitate handoff. This model is based on (1) the dissociation of Norrin from beadbound Tspan12 in the presence of saturating Fzd4 CRD (Figure 3D), and (2) a weaker measured affinity of Norrin-Tspan12LEL in the presence of saturating Fzd4 CRD (Figure 3F). We have now added wording to emphasize this in the discussion section (page 9, end of first full paragraph).

However, as you note, the Norrin-Tspan12 affinity that we measured in the presence of Fzd CRD (tens of nM) is still much stronger than the known Norrin-LRP6 affinity (0.5-1µM), which predicts that the efficiency of this handoff may be low. We have now commented on this in the discussion section and mentioned an alternative model in which Tspan12 presents the second Norrin protomer to LRP5/6 for signaling, instead of dissociating (page 9, paragraph 2). However, the handoff efficiency could also be impacted by other factors such as the relative abundance and surface distribution of Tspan12, Fzd4, LRP6 and HSPGs.

Areas that would benefit from further experiments, or a discussion, include:- The authors test a potential composite binding site of Fzd4/Tspan12 heterodimers for norrin using nanodiscs that contain on average about 1 molecule Fzd4 and 1 molecule Tspan12. The Fzd4/Tspan12 heterodimer is co-inserted into the nanodiscs supported by split-GFP tags on Fzd4 and Tspan12. The authors find no major increase in affinity, although they find changes to the Hill slope, reflecting better binding of norrin at low norrin concentrations. In 293F cells overexpressing Fzd4 and Tspan12 (which may result in a different stoichiometry) they find more pronounced effects of norrin binding to Fzd4/Tspan12. This raises the possibility that the formation of a composite binding requires Fzd4/Tspan12 complexes of higher order, for example, 2:2 Fzd4/Tspan12 complexes, where the composite binding site may involve residues of each Fzd4 and Tspan12 molecule in the complex. This could be tested in nanodiscs in which Fzd4 and Tspan12 are inserted at higher concentrations or using Fzd4 and Tspan12 that contain additional tags for oligomerization.

It is quite possible that Tspan12 and Fzd4 cluster into complexes with a stoichiometry greater than 1:1 in cells (this is supported by e.g., BRET experiments in (Ke et al., 2013)), and we mention in the discussion that that receptor clustering may be an additional mechanism by which Tspan12 exerts its function (page 10, paragraph 4). We would be quite interested to know the stoichiometry of Fzd4 and Tspan12 complexes in cells at endogenous expression levels, both in the presence and absence of Norrin, and to biochemically characterize these putative larger complexes in the future. We have amended the discussion to mention the caveat that our reconstitution experiments do not test higher-stoichiometry Fzd4/Tspan12 complexes (page 10, last paragraph).

- While Tspan12 LEL does not bind to Fzd4, the successful reconstitution of GFP from Tspan12 and Fzd4 tagged with split GFP components provides evidence for Fzd4/Tspan12 complex formation. As a negative control, e.g., Fzd5, or Tspan11 with split GFP tags (Fzd5/Tspan12 or Fzd4/Tspan11) would clarify if FZD4/Tspan12 heterodimers are an artefact of the split GFP system.

The split-GFP system allows us to co-purify receptors that do not normally co-localize (for example, as we have shown with Fzd4 and LRP6 in the absence of ligand (Bruguera et al., 2022)) so we do not mean to claim that it provides evidence for Fzd4/Tspan12 complex formation. In fact, we were unable to co-purify co-expressed Fzd4 and Tspan12 unless they were tethered with the split GFP system, and separately-purified Fzd4 and Tspan12 did not incorporate into nanodiscs together unless they were tethered by split GFP. Based on these experiments, we expect that the purported Fzd4-Tspan12 interaction that others have found by co-IP or co-localization is easily disrupted by detergent, may require a specific lipid, and/or may not be direct.

To clarify this point, we have noted in the results section that without the split GFP tags, Tspan12 and Fzd4 did not co-purify or co-reconstitute into nanodiscs, and that co-reconstitution was enabled by the split GFP system (page 6, first full paragraph).

- Fzd4/Tspan12 heterodimers stabilized by split GFP may be locked into an unfavorable orientation that does not allow for the formation of a composite binding site of FZD4 and Tspan12, this is another caveat for the interpretation that Fzd4/Tspan12 do not form a composite binding site. This is not discussed.

While the split GFP does enforce a Fzd4/Tspan12 dimer, the split GFP is removed by protease cleavage during the final step of the purification process, after the dimer is contained in a nanodisc. This should allow Fzd4 and Tspan12 to freely adopt any pose and to diffuse within the confines of the nanodisc lipid bilayer. However, it has been shown that the phospholipid bilayer in small nanodiscs is not as fluid as the physiological plasma membrane, and although we used the slightly larger belt protein (MSP1E3D1, 13 nm diameter nanodiscs), perhaps the receptors are indeed locked in some unfavorable state for this reason. Additionally, the nanodiscs are planar, so if the formation of a composite binding site requires membrane curvature, this would not be recapitulated in our system. We have cited these caveats in the discussion section (page 10, last paragraph).

- Mutations that affect the affinity of norrin/fzd4 are not used to further test if Fzd4 and Tspan12 form a composite binding site. Norrin R41E or Fzd4 M105V were previously reported to reduce norrin/frizzled4 interactions and signaling, and both interaction and signaling were restored by Tspan12 (Lai et al. 2017). Whether a Fzd4/Tspan12 heterodimer has increased affinity for Norrin R41E was not tested. Similarly, affinity of FZD4 M105V vs a Fzd4 M105V/Tspan12 heterodimer were not tested.

Since the high affinity of Norrin for both Fzd4 and Tspan12 may have obscured any enhancement of Norrin affinity for Fzd4/Tspan12 compared to either receptor alone, we did consider weakening Fzd-Norrin affinity to sensitize this experiment, inspired by the experiments you mention in (Lai et al., 2017). However, we suspected that the slight increase in Norrin affinity for the Fzd4/Tspan12 dimer compared to Fzd4 alone was driven mainly by increased avidity that enhanced binding of low Norrin concentrations, and this avidity effect would likely confound the interpretation of any experiment monitoring 2:2 complex formation. Additionally, on the basis that soluble Fzd4 extracellular domain and Tspan12 bind Norrin with negative cooperativity (Figures 3D and 3F), we concluded that this composite binding site was unlikely.

- An important conclusion of the study is that Tspan12 or Lrp6 binding to Norrin is mutually exclusive. This could be corroborated by an experiment in which LRP5/6 is inserted into nanodiscs for BLI binding tests with Norrin, or Tspan12 LEL, or a combination of both. Soluble LRP6 may remove norrin from equilibrium binding/unbinding to Tspan12, therefore presenting LRP6 in a non-soluble form may yield different results.

We agree that testing this conclusion in an orthogonal experiment would be a valuable addition to this study. We have now performed a similar experiment to the one you described, but with Norrin immobilized on biosensors, and with LRP6 in detergent competing with Tspan12 LEL for Norrin binding (Figure S12, discussed on page 8, first full paragraph). The results of this experiment show that biosensor-immobilized Norrin will bind LRP6, and that soluble Tspan12 inhibits LRP6 binding in a concentration-dependent manner. The LRP6 construct we use (residues 20-1439) includes the transmembrane domain but has a truncated C terminus, since LRP6 constructs containing the full C terminus tend to aggregate during purification. We chose to immobilize Norrin to make the experiment as interpretable as possible, since immobilizing LRP6 and competing Norrin off with the LEL could result in an increase in signal (from the LEL binding the second available Norrin protomer) as well as a decrease (from Norrin being competed off of the immobilized LRP6). We conducted the experiment in detergent (DDM) instead of nanodiscs to be able to test higher concentrations of LRP6.

- The authors use LRP6 instead of LRP5 for their experiments. Tspan12 is less effective in increasing the Norrin/Fzd4/Lrp6 signaling amplitude compared to Norrin/Fzd4/Lrp5 signaling, and human genetic evidence (FEVR) implicates LRP5, not LRP6, in Norrin/Frizzled4 signaling. The authors find that Norrin binding to LRP6 and Tspan12 is mutually exclusive, however this may not be the case for Lrp5.

This is an important point which we have now addressed in the text (page 8, end of first full paragraph). LRP5 is indeed the receptor implicated in FEVR and expressed in the relevant tissues for Tspan12/Norrin signaling. Unfortunately, LRP5 expresses poorly and we are unable to purify sufficient quantities to perform these experiments. However, LRP5 and LRP6 both transduce Tspan12-enhanced Norrin signaling in TOPFLASH assays (as you mention and as shown by (Zhou and Nathans, 2014)), bind Norrin, and are highly similar (they share 71% sequence identity overall and 73% sequence identity in the extracellular domain), so we expect their Norrin-binding sites to be conserved.

- The biochemical data are largely not correlated with functional data. The authors suggest that the Norrin R115L FEVR mutation could be due to reduced norrin binding to tspan12, but do not test if Tspan12-mediated enhancement of the norrin signaling amplitude is reduced by the R115L mutation. Similarly, the impressive restoration of binding by charge reversal mutations in site 3 is not corroborated in signaling assays.

We agree that testing the impact of Norrin mutations in cell-based signaling assays would be an informative way to further test our model. However, the Norrin mutants we tested generated poor TopFlash signals in all conditions tested. This may be due to general protein instability, weakened affinity for LRP, or weaker interactions with HSPGs. Whatever the cause, the low signal made it challenging to conclusively say whether the Norrin mutations affected Tspan12mediated signaling enhancement.

When expressed for purification, Tspan12 mutants generally expressed poorly compared to WT Tspan12, so we were concerned that differences in protein stability or trafficking would lead to lower cell-surface levels of mutant Tspan12 relative to WT in TopFlash signaling assays, which would confound interpretation of mutant Tspan’s ability to enhance Norrin signaling.

Because of these challenges, follow-up experiments to investigate the signaling capabilities of Norrin and Tspan12 mutants were not informative and we have not included them in the revised manuscript.

**Reviewer #3 (Public Review):**
Brugeuera et al present an impressive series of biochemical experiments that address the question of how Tspan12 acts to promote signaling by Norrin, a highly divergent TGF-beta family member that serves as a ligand for Fzd4 and Lrp5/6 to promote canonical Wnt signaling during CNS (and especially retinal) vascular development. The present study is distinguished from those of the past 15 years by its quantitative precision and its high-quality analyses of concentration dependencies, its use of well-characterized nano-disc-incorporated membrane proteins and various soluble binding partners, and its use of structure prediction (by AlphaFold) to guide experiments. The authors start by measuring the binding affinity of Norrin to Tspan12 in nanodiscs (~10 nM), and they then model this interaction with AlphaFold and test the predicted interface with various charge and size swap mutations. The test suggests that the prediction is approximately correct, but in one region (site 1) the experimental data do not support the model. [As noted by the authors, a failure of swap mutations to support a docking model is open to various interpretations. As AlphFold docking predictions come increasingly into common use, the compendium of mutational tests and their interpretations will become an important object of study.] Next, the authors show that Tspan12 and Fzd4 can simultaneously bind Norrin, with modest negative cooperativity, and that together they enhance Norrin capture by cells expressing both Tspan12 and Fzd4 compared to Fzd4 alone, an effect that is most pronounced at low Norrin concentration. Similarly, at low Norrin concentration (~1 nM), signaling is substantially enhanced by Tspan12. By contrast, the authors show that LRP6 competes with Tspan12 for Norrin binding, implying a hand-off of Norrin from a Tspan12+Fzd4+Norrin complex to a LRP5/6+Fzd4+Norrin complex. Thanks to the authors' careful dose-response analyses, they observed that Norrin-induced signaling and Tspan12 enhancement of signaling both have bell-shaped dose-response curves, with strong inhibition at higher levels of Norrin or Tspan12. The implication is that the signaling system has been built for optimal detection of low concentrations of Norrin (most likely the situation in vivo), and that excess Tspan12 can titrate Norrin at the expense of LRP5/6 binding (i.e., reduction in the formation of the LRP5/6+Fzd4+Norrin signaling complex). In the view of this reviewer, the present work represents a foundational advance in understanding Norrin signaling and the role of Tspan12. It will also serve as an important point of comparison for thinking about signaling complexes in other ligand-receptor systems.
**Recommendations for the authors:**

**Reviewer #2 (Recommendations For The Authors):**
- In Figure 5F high concentrations of transfected Tspan12 plasmid inhibit signaling, which the authors interpret to support the model that Tspan12/Norrin binding prevents Norrin/LRP6/FZD4 complex formation. Alternatively, the cells do not tolerate the expression of the tetraspanin at high levels, for example, due to misfolding and aggregate formation. To distinguish these possibilities: Do high levels of Tspan12 overexpression also inhibit signaling induced by Wnt3a and appropriate Frizzled receptors, even though Tspan12 has no influence on Wnt/LRP6 binding?

We thank the reviewer for suggesting this important control experiment. We have added the Wnt-simulated TOPFLASH values to the figure in 5F for all conditions. In repeating this experiment, we noticed that high levels of transfected Tspan12 may decrease cell viability and therefore have adjusted the range of transfected Tspan12 in the new Figure 5F (discussed on page 8, second full paragraph). Under this new protocol, both Norrin- and Wnt-stimulated signaling were inhibited by the highest amount of transfected Tspan12. However, Norrinstimulated signaling is inhibited by lower amounts of transfected Tspan12 than Wnt-stimulated signaling, and to a greater extent, supporting our proposed model that Tspan12 competes with LRP for Norrin binding.

- Is Tspan12 with c-terminal rho-tag (the form incorporated into nanodiscs) also used for functional luciferase assays, or was untagged Tspan12 used for the luciferase assays in Fig 4D and 5F? Does the c-terminal tag interfere with Tspan12-mediated enhancement of Norrin signaling?

For the luciferase assays included in this manuscript, wildtype, full-length, untagged Tspan12 is used. We have clarified this in our methods section. When we tested the wildtype vs Cterminally rho1D4-tagged version of Tspan12 in TOPFLASH assays, we saw that the enhancement of Norrin signaling by Tspan12-1D4 was weaker than enhancement by untagged Tspan12. This is consistent with the finding reported in Cell Reports (Lai et al., 2017) that a chimeric Tspan12 receptor with its C-terminus replaced with that of Tspan11 was still capable of enhancing Norrin signaling, though to a lesser extent than WT Tspan12. The deficiency of signaling by our rho1D4-tagged Tspan12 could be due to a difference in receptor expression level or trafficking, but in the absence of a reliable antibody against Tspan12, we were unable to assess the expression levels or localization of the untagged Tspan12 to compare it to the rho1D4-tagged version. (For binding experiments, we reasoned that the C-terminal tag should not affect Tspan12’s ability to bind Norrin extracellularly, especially as we found that purified fulllength Tspan12 and Tspan12∆C (residues 1-252) bound Norrin equally well; we have added this comparison to table S1.)

**Reviewer #3 (Recommendations For The Authors):**
Minor comments.Based on the Fzd4-Dvl binding experiment, the authors might state explicitly the possibility that Tspan12's relevance is entirely accounted for by extracellular ligand capture.

We have stated this possibility explicitly in the discussion section (page 9, last paragraph).

Page 4, 3rd paragraph. I suggest "To experimentally test this structural prediction..." rather than "validate".

Thank you for this suggestion; we have replaced this wording.

This next item is optional, but I hope that the authors will consider it. This manuscript provides an opportunity for the authors to be more expansive in their thinking, and to put their work into the larger context of ligand+receptor+accessory protein interactions. The authors describe the Wnt7a/7b-Gpr124-RECK system and the role of HSPs in Norrin and Wnt signaling, but perhaps they can also comment on non-Wnt ligand-receptor systems where accessory proteins are found. They might add a figure (or supplemental figure) with a schematic showing the roles of HSP and Gpr124-RECK, and some non-Wnt ligand-receptor systems. This would help to make the present work more widely influential.

Thank you for this suggestion. We have added a figure (Figure 6, discussed on page 10, paragraphs 2 and 3) and expanded our discussion to include other co-receptor systems. We have specifically focused on co-receptors that both capture ligands and interact with their primary receptor(s), thus delivering ligands to their receptors, as we have proposed for Tspan12. Within Wnt signaling, other co-receptor systems with this mechanism are RECK/Gpr124 (for Wnt7a/b) and Glypican-3. We found it interesting that this mechanism is also shared by several growth factor pathways with cystine knot ligands (like Norrin), so we have illustrated and mentioned three of these examples.